# Mitigating Forgetting Between Supervised and Reinforcement Learning Yields Stronger Reasoners

## Abstract

Large Language Models (LLMs) show strong reasoning abilities, often amplified by Chain-of-Thought (CoT) prompting and reinforcement learning (RL). Although RL algorithms can substantially improve reasoning, they struggle to expand reasoning boundaries because they learn from their own reasoning trajectories rather than acquiring external knowledge. Supervised fine-tuning (SFT) offers complementary benefits but typically requires large-scale data and risks overfitting. Recent attempts to combine SFT and RL face three main challenges: data inefficiency, algorithm-specific designs, and catastrophic forgetting. We propose a plug-and-play framework that dynamically integrates SFT into RL by selecting challenging examples for SFT. This approach reduces SFT data requirements and remains agnostic to the choice of RL or SFT algorithm. To mitigate catastrophic forgetting of RL-acquired skills during SFT, we select high-entropy tokens for loss calculation and freeze parameters identified as critical for RL. Our method achieves state-of-the-art (SOTA) reasoning performance using only 1.5% of the SFT data and 20.4% of the RL data used by prior SOTA, providing an efficient and plug-and-play solution for combining SFT and RL in reasoning post-training.

## 1 Introduction

Recent Large Language Models (LLMs) has shown reasoning capability (Jaech et al., 2024; Guo et al., 2025; Anthropic, 2025). The reasoning capability are highly dependent on the use of the Chain-of-Thought (CoT) thinking pattern trained by supervise fine-tuning (SFT) or reinforcement learning (RL). Although popular RL algorithms such as PPO (Schulman et al., 2017), GRPO (Guo et al., 2025), and DAPO (Yu et al., 2025) are promising in multiple reasoning tasks, recent studies argue that RL training does not truly extend a model's reasoning boundaries. Because RL trains the LLMs based on self-generated rollout samples, RL primarily reshapes the model's internal probability distribution rather than enabling the acquisition of new knowledge (Yue et al., 2025; Wang et al., 2025b). Compared with the weaknesses of RL, SFT can import external out-of-distribution samples into the model, although it carries risks of memorization or overfitting (Chu et al., 2025) and reduced generalization (Yuan et al., 2025).

Consequently, except for the common recipe (Yang et al., 2024; Shao et al., 2024) that uses SFT then RL to post-train the model, many recent studies with better reasoning performances (Chen et al., 2025; Yan et al., 2025) explore how to leverage the strengths of SFT and RL while avoiding their drawbacks by strategically combining them into an interconnected RL–SFT optimization process to balance knowledge injection with self-exploration. However, these methods face one or more of the following three significant challenges: *(i)* Dependence on large amounts of high-quality SFT data. LUFFY (Yan et al., 2025) and SRFT (Fu et al., 2025) require extensive high-quality SFT answers to make the actor model imitate stronger offline teacher models or annotators. *(ii)* Algorithm-specific designs. Yan et al. (2025); Fu et al. (2025); Liu et al. (2025b) are tailored to particular RL formulations to incorporate SFT supervision. ReLIFT (Ma et al., 2025) avoids the first two limitations by interleaving SFT and RL, but it still suffers from the following challenge, which largely degrades the model's reasoning capability. *(iii)* Lack of handling catastrophic forgetting between SFT and RL. Our analysis indicates that SFT induces catastrophic forgetting when jointly optimized with RL, and none of the current methods have proposed a corresponding solution.

In light of this, we propose a plug-and-play framework, **MIFO** (**MI**tigating **FO**rgetting Between SFT and RL), which integrates SFT with RL while addressing the above issues **simultaneously**. To tackle *(i)* the need for large amounts of SFT data and *(ii)* algorithm-specific designs, we dynamically interleave SFT within RL and select SFT examples based on rollout accuracy, while selecting tokens for the SFT loss according to entropy. This dynamic selection ensures that the actor model uses only the minimal necessary SFT data to acquire out-of-distribution reasoning knowledge. Because interleaved SFT and RL are not merged into a single optimization objective, our approach can be seamlessly applied to new RL or SFT algorithms.

Most importantly, MIFO is designed with core principle to solve *(iii)* catastrophic forgetting. Our evidence shows that forgetting arises when extensive SFT updates overwrite the information acquired through RL. We therefore design MIFO with two complementary mechanisms. *First*, the aforementioned data and token selection strategy not only enables MIFO plug-and-play with reduced data usage, but also limits the magnitude of SFT updates, which decreases the forgetting of the knowledge learned by previous RL. *Second*, our analysis reveals an asymmetry between SFT and RL parameter updates. SFT tends to update parameters redundantly, so removing part of its updates does not harm performance significantly. RL, on the other hand, updates parameters in a more parsimonious way, and omitting part of its updates leads to clear performance degradation. Based on this observation, MIFO dynamically identifies RL-critical parameters, freezes them during SFT, and unfreezes them in the subsequent RL step. This design protects important RL parameter updates from being overwritten by SFT, effectively mitigating catastrophic forgetting.

The contributions of this work are summarized below:

- We introduce **MIFO**, a plug-and-play framework that simultaneously addresses data hunger, algorithm-specific design, and catastrophic forgetting problems, providing an effective and efficient solution for combining SFT and RL in reasoning post-training.

- **MIFO** consists of two components: data processing and parameter manipulation. We sample rollouts to select only the necessary questions for SFT. Based on our analysis of SFT and RL update patterns, we maintain a map of RL-sensitive parameters, freezing them during SFT and unfreezing them for RL to promote long-term training stability.

- **MIFO** achieves SOTA reasoning performance with substantially less SFT and RL data, and is able to be adapted to new RL–SFT combinations with different algorithms. MIFO delivers high reasoning efficiency, with average response length comparable to strong baselines.

## 2 RELATED WORKS

**RL for Reasoning.** RL emerges as an important paradigm to incentivize LLMs' reasoning capability (Jaech et al., 2024; Shao et al., 2024; Team et al., 2025) since SFT is identified having the drawback of memorization effect (Chu et al., 2025; Yuan et al., 2025). Popular approaches like PPO (Schulman et al., 2017), GRPO (Guo et al., 2025), Dr.GRPO (Liu et al., 2025c) and DAPO (Yu et al., 2025) have shown the great improvement for reasoning tasks. Recent research also tends to question how RL works. Yue et al. (2025) claims that RL doesn't expand the boundary of reasoning capability but lets the right answer emerge early for multiple tryouts. Wang et al. (2025b) discovers that the main role of RL is to incentivize the knowledge pre-trained to the model. In light of this, our work aims to balance RL and SFT to learn the knowledge and then incentivize reasoning capability.

**Interplaying SFT and RL.** Combining SFT with RL has emerged as an effective way for boosting LLM reasoning. Some methods interleave SFT with RL using demonstrations mined during training (Ma et al., 2025), or imitate off-policy traces from stronger teachers (Yan et al., 2025; Fu et al., 2025); others balance the two signals via schedules (SASR) (Chen et al., 2025), prompt-level supervision during rollouts (Liu et al., 2025a), or joint loss formulations (SuperRL) (Liu et al., 2025b). While successful, these approaches typically require substantial supervised data and/or algorithm-specific integration. Our design achieves great gains with much less training data and minimal coupling, and is closest in spirit to ReLIFT's interleaving strategy (Ma et al., 2025).

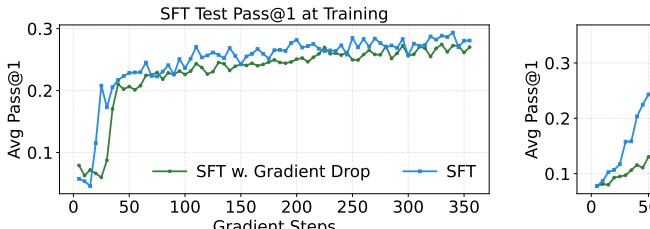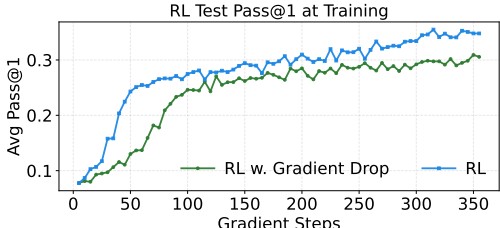

Figure 1: Dropping gradients when updating parameters causes more performance drop on RL.

## 3   CONTRASTING THE ROLES OF SFT AND RL IN REASONING TRAINING

We outline two key insights about LLM weights updates under SFT and RL. **First**, SFT modifies parameters with redundancy; even if updates of some parameters are randomly removed, the model retains comparable performance. RL, in contrast, updates parameters in a more parsimonious manner, and removing even part of its updates leads to clear performance degradation. **Second**, combining SFT and RL during post-training introduces the risk of catastrophic forgetting, especially when SFT follows RL. We hypothesize that this occurs because SFT applies much larger updates than RL, thereby overwriting information learned during RL.

### 3.1   SFT REDUNDANT, RL PARSIMONIOUS

Two experiments comprehensively provide the conclusion that SFT has more redundancy in parameter updating compared to RL, from gradient to parameter perspectives. We set the same learning rate and data amount, epochs = 1, and other configurations are the same as in Appendix D. We study two sparsification regimes on model updating by observing model performance (average pass@1 on 6 reasoning test sets). *(i) Online gradient sparsification.* Let $p_{on} \in [0, 1]$ denote the drop rate per step. At each gradient step $t$, we have model parameters $\theta_t \in \mathbb{R}^d$ with loss gradient $g_t = \nabla_\theta \mathcal{L}(\theta_t)$. We draw an i.i.d. coordinate mask $m_t \sim \text{Bernoulli}(1 - p_{on})^d$ (1=*keep*, 0=*drop*) and update with the masked gradient

$$\tilde{g}_t \;=\; m_t \odot g_t, \qquad \theta_{t+1} \;=\; \text{AdamW}(\theta_t, \tilde{g}_t),$$

where $\text{AdamW}$ represents one AdamW optimization step. In our runs we set $p_{on} = 0.5$, i.e., randomly drop 50 percent of gradient coordinates each step, independently across steps and coordinates. *(ii) Post-hoc update sparsification.* Let $p_{post} \in [0, 1]$ denote the pruning rate applied once after training. First train unperturbed for $T$ steps to obtain $\theta_0 \to \theta_T$, define the net update $\Delta\theta = \theta_T - \theta_0$, then sample a single mask $m \sim \text{Bernoulli}(1 - p_{post})^d$ and form

$$\theta_T^{post} \;=\; \theta_0 \;+\; m \odot \Delta\theta,$$

i.e., randomly prune a $p_{post}$-fraction of the total parameter change coordinatewise. Both regimes are instantiated identically for SFT and RL (only their loss $\mathcal{L}$ differs).

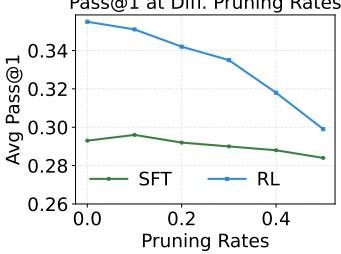

Figure 2: Compared w. SFT, RL has a notable drop when the pruning rate $p_{post}$ increases.

Both experiments indicate that SFT produces more redundant parameter updates, whereas RL induces more parsimonious updates. Dropping a portion of SFT gradients or parameter updates yields comparable performance, but doing so for RL leads to clear degradation. *(i) Online gradient sparsification.* Figure 1 shows that during training, the SFT w. Gradient Drop model matches the SFT baseline after roughly the 40th step. By contrast, the RL w. Gradient Drop model suffers a larger performance drop relative to the RL baseline, and the gap only begins to close around the 110th step. At convergence, the average pass@1 (percentage points) gap is 0.7 for SFT versus 4.23 for RL . This highlights SFT's higher redundancy compared with RL. *(ii) Post-hoc update sparsification.* Figure 2 varies the pruning rate from 0 to 0.5. For RL, the average pass@1 drops consistently from 35.5 to 29.9 (-5.6). For SFT, accuracy initially increases from 29.3 to 29.6 (+0.3) at pruning rate 0.1, then gradually decreases to 28.4 (-0.9). These results

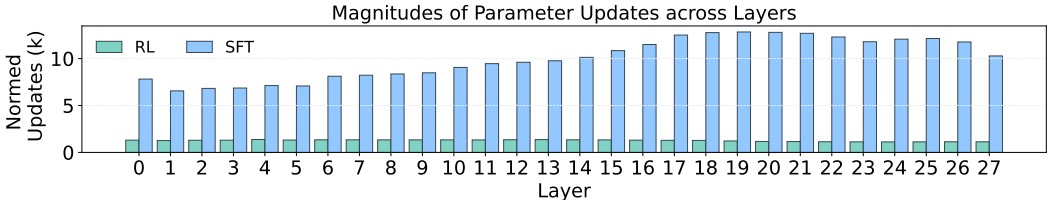

Figure 3: SFT induces much more parameter updating on magnitude, compared with RL.

suggest that SFT exhibits both redundancy and overfitting (a small amount of pruning improves performance), whereas RL updates are more parsimonious, with higher prune rates steadily harming performance. We further provide explanations on the reason for these experiment designs in Appendix F.1, and extra experiment in Appendix E.4. To better understand this phenomenon, we also provide a corresponding theoretical analysis and proof in Appendix C.

## 3.2 SFT FORGETS RL

Recent SOTA reasoning post-training pipelines, such as DeepSeekMath (Shao et al., 2024), LUFFY (Yan et al., 2025), ReLIFT (Ma et al., 2025), and SRFT (Fu et al., 2025), integrate SFT and RL, either explicitly or implicitly. However, we identify that within these combinations, the SFT component induces the forgetting of information previously learned via RL component. We quantitatively measure this phenomenon by first training a model for 50 steps using RL, and subsequently shifting to these pipelines' training objectives containing SFT. As shown in Figure 4, with the exception of our MIFO framework, all other methods exhibit a degradation in reasoning performance, indicating the occurrence of forgetting.

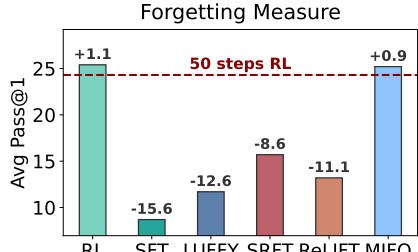

Figure 4: Average test pass@1 for measuring forgetting quantitatively.

We hypothesize that this degradation occurs because SFT induces much larger magnitudes of parameter updates, thereby overwriting RL-induced updates and triggering catastrophic forgetting of knowledge learned by RL. This hypothesis is supported by the following observation. Using the same configuration as Section 5.1 (with epochs set to 1), we measure the **magnitude of parameter updates** for SFT or RL from the start step 0 to the end of training in step $T$, defined as $\Delta\theta = \|\theta_T - \theta_0\|_2$. Figure 3 shows that across all layers, SFT produces much larger parameter changes than RL, which can overwrite RL updates and cause catastrophic forgetting of RL-acquired information. The forgetting likely stems from the heterogeneity of data and training objectives between SFT and RL because of larger SFT updates.

Recent work has combined SFT and RL within a single optimization procedure and achieved SOTA performance (Section 2). While the idea of dynamically balancing learning from external distributions via SFT and self-exploration via RL is compelling, Figure 3 suggests that the full potential of SFT and RL remains untapped because prior methods did not explicitly mitigate this forgetting risk.

## 3.3 MOTIVATION

When SFT overwrites RL due to excessive updates, can we reduce its update magnitude to mitigate this forgetting? Section 3.1 shows that SFT exhibits substantial redundancy in parameter updates relative to RL and carries a notable risk of overfitting. Our motivation follows directly: *We combine SFT with RL for reasoning post-training while constraining SFT-induced parameter updates to reduce catastrophic forgetting on the information learned by RL.*

## 4 METHOD

Based on the insights in Section 3, we propose **MIFO** as illustrated in Figure 5, which incorporates interleaved RL and SFT (multiple connected RL→ SFT intervals). The central principle is to limit

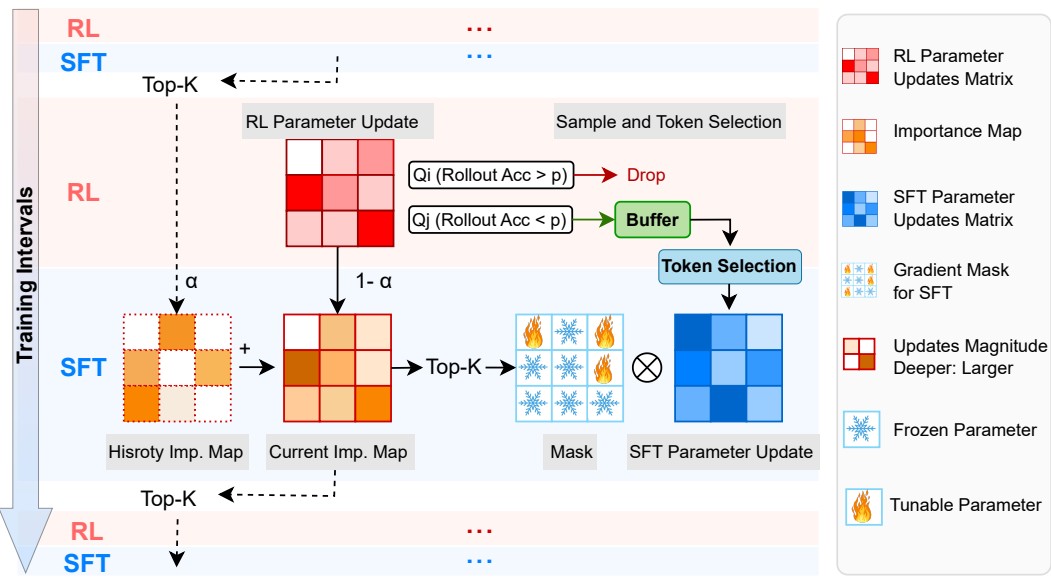

Figure 5: MIFO is a training pipeline with multiple connected RL→ SFT intervals. In **RL** training: it selects data for SFT and decides the important RL parameters at the end; In **SFT** training: RL-important parameters are frozen, and only high-entropy tokens are used for loss calculation.

SFT parameter updates, thereby reserving parameter space for the more parsimonious updates of RL. Guided by this, MIFO introduces two key designs: *Data processing*. At the sample level, we use RL rollouts to identify questions the model currently struggles with, and we curate an online SFT set containing only these cases with verified solutions. At the token level, we focus the SFT objective on the most informative, high-uncertainty tokens rather than spreading gradients over easy tokens. *Parameter freezing*. To directly reduce forgetting between stages, we track which parameters were most updated during RL and temporarily freeze these RL-critical components during subsequent SFT, then unfreeze them before the next RL step.

## 4.1 DATA PROCESSING

**Interleaving SFT and RL with Buffer.** We adopt GRPO (can be changed to another RL algorithm) for RL training, and select challenging data samples according to the GRPO rollout accuracy for the following SFT training. Let the policy before and after the update be $\pi_{\theta_{old}}$ and $\pi_\theta$. Given a question $q$, a set of solutions $\{o_i\}_{i=1}^N$ generated by $\pi_{\theta_{old}}$, and a reward function $R(\cdot)$, the objective is

$$\mathcal{L}_{\text{GRPO}}(\theta) = \frac{1}{\sum_{i=1}^N |o_i|} \sum_{i=1}^N \sum_{t=1}^{|o_i|} \min\left[r_{i,t}(\theta)\, A_i,\ \text{clip}\left(r_{i,t}(\theta),\, 1-\epsilon,\, 1+\epsilon\right) A_i\right],$$

where $r_{i,t}(\theta) = \frac{\pi_\theta(o_{i,t}|q,o_i<t)}{\pi_{\theta_{old}}(o_{i,t}|q,o_i<t)}, A_i = \frac{R(o_i)-\text{mean}\left(\{R(o_j)\}_{j=1}^N\right)}{\text{std}\left(\{R(o_j)\}_{j=1}^N\right)}$. For each question $q$, GRPO produces $N$ solutions and yields an empirical accuracy $\text{acc}(q)$. We then collect questions with $\text{acc}(q) \leq p$. For these challenging cases, where the policy exhibits a low success rate under self-exploration, we assume that additional out-of-distribution knowledge is required. We obtain high-quality solutions from a stronger reasoning model or from human experts, and insert them into an SFT buffer $\text{Buffer}_{\text{FT}}$:

$$\text{Buffer} = \left\{(q,s)\ \middle|\ \text{acc}(q) \leq p,\ s = D(q),\ \text{extract}(s) = a\right\}, \qquad \text{Buffer}_{\text{FT}} \leftarrow \text{Buffer}_{\text{FT}} \cup \text{Buffer},$$

where $D(q)$ denotes a CoT solution obtained either offline or online from an external model or a human annotator, $\text{extract}(s)$ is the final answer extracted from $s$, and $a$ is the ground-truth answer for $q$. We retain only pairs satisfying $\text{extract}(s) = a$. Once the buffer size exceeds a preset threshold $S$ (i.e., $|\text{Buffer}_{\text{FT}}| \geq S$), training shifts from RL to SFT. Then, the training will shift from SFT to RL when samples in buffer are all trained and the buffer is emptied.

ReLIFT (Ma et al., 2025) first proposed using SFT buffer, while it is restricted to $\mathrm{acc}(q) = 0$. We generalize this by using a threshold $\mathrm{acc}(q) \leq p$, which admits more informative yet solvable cases. The contribution is not the threshold alone. It indicates that our subsequent modules explicitly mitigate forgetting to enlarge the usable data pool, and maximize the learning on existing knowledge of data to improve reasoning capability. Detailed discussion is provided in Appendix F.3.

**SFT with High-entropy Tokens.** After RL, MIFO SFT the model only on high-entropy tokens from $\mathrm{Buffer}_{\mathrm{FT}}$ to limit the magnitude of parameter updates. Entropy (Shannon, 1948) describes the uncertainty, and high-entropy tokens mark positions of model uncertainty. Concentrating learning on these tokens encourages acquisition of missing knowledge while avoiding unnecessary fitting of low-entropy (confident) tokens, which can exacerbate overfitting or lead to entropy collapse. For each token position $t$ in a response, define the entropy $H_t = -\sum_{j=1}^{V} p_{t,j} \log p_{t,j}$, where $\mathbf{p}_t = \pi_\theta(\cdot \mid q, s_{<t})$, and $V$ is the vocabulary size. We then compute an example-specific threshold $\tau_\rho(q, s)$ such that the top $\rho \in (0, 1]$ fraction of tokens by entropy in $(q, s)$ satisfy $H_t \geq \tau_\rho(q, s)$. The SFT objective minimizes loss only over these high-entropy tokens:

$$\mathcal{L}_{\mathrm{SFT}}(\theta) = -\frac{1}{|s|} \sum_{t=1}^{|s|} \mathbb{I}[H_t \geq \tau_\rho(q, s)] \log \pi_\theta(s_t \mid q, s_{<t}), \qquad (q, s) \in \mathrm{Buffer}_{\mathrm{FT}}.$$

### 4.2 PARAMETER FREEZING

The second component of **MIFO** monitors RL-induced parameter updates, and freezes the most updated (important) parameters during RL for the subsequent SFT phase, thereby preserving RL-acquired knowledge and mitigating catastrophic forgetting caused by SFT. Let $\boldsymbol{\theta}_i = \{\theta_i^1, \ldots, \theta_i^d\}$ denote the model parameters ($d$ is the number of named parameters) in the $i$-th RL→SFT interval. To reduce forgetting, we maintain a history importance map $\mathbf{C}_i \in \mathbb{R}^d$ that tracks important RL updates, and we freeze parameters with the largest importance scores for the following SFT. Let $\boldsymbol{\theta}_{\mathrm{RL,start},i}$ and $\boldsymbol{\theta}_{\mathrm{RL,end},i}$ be the parameters at the start and end of the RL session in interval $i$. Define the RL update $\Delta\boldsymbol{\theta}_{\mathrm{RL},i}$ by

$$\Delta\boldsymbol{\theta}_{\mathrm{RL},i}^j = \left\| \theta_{\mathrm{RL,end},i}^j - \theta_{\mathrm{RL,start},i}^j \right\|_2, \quad j = 1, \ldots, d.$$

Initialize $\mathbf{C}_0 = \mathbf{0}$. Let $\alpha \in [0, 1)$ be the history coefficient and define the current importance map

$$\tilde{\mathbf{C}}_i = \alpha\, \mathbf{C}_{i-1} + (1 - \alpha)\, \Delta\boldsymbol{\theta}_{\mathrm{RL},i}.$$

Here $\mathbf{C}$ serves as history importance map for RL parameter updates, with global information along the training. We then select the top-$k$ entries by magnitude to find the most-learned (important) parameters during RL training, $\mathcal{I}_i = \mathrm{TopK}(\tilde{\mathbf{C}}_i, k)$, and define a binary mask $\mathbf{M}_i \in \{0, 1\}^d$ with

$$\mathbf{M}_i^j = \mathbf{1}[j \in \mathcal{I}_i], \qquad \mathbf{C}_i = \mathbf{M}_i \odot \tilde{\mathbf{C}}_i,$$

where $\odot$ denotes element-wise multiplication. The masked $\mathbf{C}_i$ is passed to the next RL→SFT interval as history. To protect against catastrophic forgetting in the subsequent SFT phase, we freeze RL-important parameters indexed by $\mathcal{I}_i$ by zeroing their SFT gradients:

$$\nabla_{\boldsymbol{\theta}_i} \mathcal{L}_{\mathrm{SFT}} \leftarrow (\mathbf{1} - \mathbf{M}_i) \odot \nabla_{\boldsymbol{\theta}_i} \mathcal{L}_{\mathrm{SFT}}.$$

After SFT completes, all parameters are unfrozen for the next RL session. RL updates produce smaller parsimonious updates, which almost do not induce forgetting of SFT-acquired knowledge. This RL→SFT cycle repeats throughout post-training, promoting long-term training stability. When $\alpha = 0$, we denote the variant as **MIFO**$^\dagger$, meaning no historical RL-importance is carried across intervals. Pseudocode of MIFO is provided in Algorithm 1.

## 5 EXPERIMENT

### 5.1 SETUPS

**Evaluation.** We evaluate on five widely used mathematical reasoning benchmarks: AIME 2024, AIME 2025, AMC (Li et al., 2024), OlympiadBench (He et al., 2024), and MATH500 (Hendrycks

Table 1: Overall reasoning performances ↑ and average response length (#token) ↓ on **Qwen2.5-Math-1.5B**. **Bold** and underline indicate the best and second-best results, respectively.

| Model | AIME-24 | AIME-25 | AMC | MATH-500 | Olympiad | MMLU-Pro | Average | Length |
|---|---|---|---|---|---|---|---|---|
| *Base* | | | | | | | | |
| **Qwen-Math** | 3.0 | 1.4 | 19.4 | 44.4 | 16.7 | 5.0 | 15.0 | 4043 |
| **Qwen-Math-Instruct** | 8.1 | 6.6 | 36.9 | 66.0 | 30.7 | 24.2 | 28.8 | 5806 |
| *Supervised Fine-tuning* | | | | | | | | |
| **SFT** | 12.7 | 13.0 | 40.9 | 71.8 | 33.8 | 23.5 | 32.6 | 13403 |
| *Reinforcement Learning* | | | | | | | | |
| **RL** | 10.2 | 8.7 | 44.7 | 71.4 | 34.5 | 28.7 | 33.0 | **913** |
| **Oat-Zero** | 17.8 | 10.8 | 47.2 | 73.0 | 35.8 | 20.9 | 34.3 | 2605 |
| **OpenReasoner** | 3.2 | 1.3 | 27.6 | 55.2 | 24.0 | 27.9 | 23.2 | 2445 |
| *SFT+RL* | | | | | | | | |
| **SFT→RL** | 12.5 | 9.2 | 43.4 | 72.0 | 34.9 | 29.0 | 33.5 | 12601 |
| **ReLIFT** | 13.1 | 8.8 | 42.5 | 73.6 | 36.0 | 30.1 | 34.0 | 4421 |
| **LUFFY** | 15.9 | **13.1** | 46.3 | **80.0** | 41.0 | 34.2 | 38.4 | 3406 |
| **MIFO**[†] | 19.0 | 12.6 | 49.1 | 78.0 | **43.9** | **36.2** | 39.8 | 985 |
| **MIFO** | **19.2** | 12.0 | **50.3** | 78.8 | 43.3 | 36.1 | **40.0** | 2518 |

Table 2: Overall reasoning performances ↑ and average response length (#token) ↓ on **Qwen2.5-Math-7B**. **Bold** and underline indicate the best and second-best results, respectively.

| Model | AIME-24 | AIME-25 | AMC | MATH-500 | Olympiad | MMLU-Pro | Average | Length |
|---|---|---|---|---|---|---|---|---|
| *Base* | | | | | | | | |
| **Qwen-Math** | 5.1 | 1.4 | 22.6 | 37.2 | 19.0 | 30.5 | 19.3 | 2408 |
| **Qwen-Math-Instruct** | 9.7 | 8.2 | 39.9 | 77.2 | 34.1 | 33.0 | 33.7 | 14179 |
| *Supervised Fine-tuning* | | | | | | | | |
| **SFT** | 26.9 | 23.1 | 58.9 | 85.0 | 50.7 | 48.6 | 48.9 | 10166 |
| *Reinforcement Learning* | | | | | | | | |
| **RL** | 22.1 | 13.9 | 59.5 | 84.2 | 44.7 | 48.3 | 45.5 | **653** |
| **Oat-Zero** | 33.4 | 11.9 | 61.2 | 78.0 | 43.4 | 41.7 | 44.9 | 1767 |
| **OpenReasoner** | 16.5 | 15.0 | 52.1 | 82.4 | 47.1 | 58.7 | 45.3 | 1652 |
| *SFT+RL* | | | | | | | | |
| **SFT→RL** | 29.6 | 19.6 | 61.1 | 84.2 | 51.6 | 49.6 | 49.3 | 10832 |
| **ReLIFT** | 27.2 | 20.1 | 64.9 | 85.2 | 53.6 | 52.5 | 50.6 | 1299 |
| **LUFFY** | 27.2 | 18.1 | 61.1 | 85.6 | 53.6 | 52.6 | 49.7 | 1762 |
| **SRFT** | **33.4** | 23.1 | 65.6 | 88.6 | 57.9 | **55.8** | 54.1 | 1112 |
| **MIFO**[†] | 28.2 | **28.9** | 65.6 | **89.0** | 56.4 | 54.0 | 53.7 | 1371 |
| **MIFO** | 31.7 | 26.0 | **66.2** | 87.8 | **59.4** | 54.8 | **54.3** | 1067 |

et al., 2021). For the smaller test sets (AIME 2024, AIME 2025, and AMC), we report *avg@32* (*pass@1* for Section 3). For the larger test sets (OlympiadBench and MATH500), we report *pass@1*. We also include MMLU-Pro (Wang et al., 2024) to assess the model's generalized reasoning ability.

**Datasets.** For the training set, we utilize the version of the dataset OpenR1-Math-46k from Yan et al. (2025), which contains a subset of OpenR1-Math-220k (Face, 2025), with prompts from NuminaMath (Li et al., 2024) in 45.8k prompts and their CoT demonstrations.

**Baselines.** We compare against four categories of baselines. (i) **Base models**: **Qwen-Math-2.5** and **Qwen-Math-2.5-Instruct**. (ii) **SFT-only**: **SFT**, a supervised fine-tuned variant of Qwen-Math-2.5. (iii) **RL-only**: **RL**, which is trained by vanilla GRPO; **Oat-Zero** (Liu et al., 2025c), which uses GRPO with variance removal in advantage computation and token-level normalization; **PRIME-Zero** (Cui et al., 2025), which employs implicit process rewards; **OpenReasonerZero** (Hu et al., 2025), which uses rule-based rewards. (iv) **Joint SFT+RL**: **SFT→RL**, which first applies SFT then runs RL; **LUFFY** (Yan et al., 2025), which integrates on-policy rollouts with off-policy reasoning traces within RL training. **SRFT**(Fu et al., 2025), a single-stage method that unifies both fine-tuning paradigms through weighting mechanisms (only the 7B model accessible).

We provide other implementation details in Appendix D.

Table 3: Ablation study on Qwen2.5-Math-1.5B model. **Bold** and underline indicate the best and second-best results, respectively.

| Model | AIME-24 | AIME-25 | AMC | MATH-500 | Olympiad | MMLU-Pro | Average | Length |
|---|---|---|---|---|---|---|---|---|
| **Qwen2.5-Math** | 3.0 | 1.4 | 19.4 | 44.4 | 16.7 | 5.0 | 15.0 | 4043 |
| + Interleave | 13.1 | 8.8 | 42.5 | 73.6 | 36.0 | 30.1 | 34.0 | 4421 |
| + Interleave + ES | 16.6 | 12.5 | 49.1 | **79.4** | 42.3 | 35.2 | 39.2 | 3246 |
| + Interleave + PF | 15.7 | **14.5** | 45.8 | 77.0 | 41.7 | 35.9 | 38.4 | 4490 |
| **MIFO** | **19.2** | 12.0 | **50.3** | 78.8 | **43.3** | **36.1** | **40.0** | **2518** |

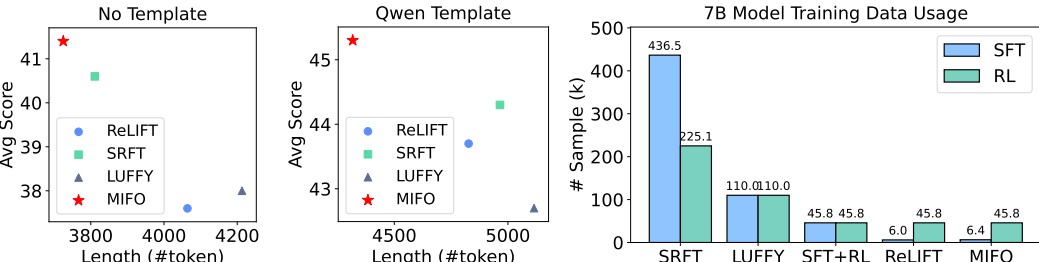

Figure 6: Average reasoning score vs. response lengths with no template (**left**) and Qwen template (**middle**) for 7B model; SFT and RL data usage for training 7B model (**right**).

## 5.2 MAIN RESULTS

Tables 1 and 2 present the main results regarding reasoning performance, response efficiency, and training data efficiency for baselines, our method **MIFO**, and **MIFO**[†]. **MIFO**[†] is a special case when the history coefficient $\alpha$ is set to 0.

**Reasoning Performance.** Tables 1 and 2 show that across five competition-level benchmarks and one out-of-distribution benchmark, our method consistently attains the best or second-best results. On the 1.5B model, it yields large gains on AMC (+4.0 over LUFFY) and OlympiadBench (+7.9 over ReLIFT). On the 7B model, it substantially outperforms SRFT on AIME 2025 (+5.8) and OlympiadBench (+1.5). Overall, our approach achieves the best average reasoning performance for both 1.5B and 7B models, highlighting the effectiveness of MIFO.

**Data Efficiency.** Because of the introduced data processing and parameter freezing design, MIFO maximizes the utility of existing data and thus reduces data requirements. Figure 6 (right) compares data usage among joint SFT+RL methods. MIFO uses only 1.5% of the SFT data and 20.4% of the RL data required by the prior SOTA SRFT, and 5.8% of the SFT data and 41.6% of the RL data used by LUFFY. MIFO employs a similar data budget compared to ReLIFT, and achieves notably better performance. More details and comparison for 1.5B model is listed in Appendix D.

**Response Length Efficiency.** Tables 1 and 2 show that MIFO yields more concise reasoning traces beyond accuracy. On the 1.5B model, MIFO and MIFO[†] produce outputs with average lengths of 2,518 and 913 tokens, respectively, shorter than all baselines except the RL-only model. On the 7B model, the gap is even more pronounced: MIFO and MIFO[†] average 1,067 and 1,371 tokens, respectively, with only the RL-only model shorter at 653 tokens. These results indicate that MIFO improves reasoning while also producing more efficient responses with less tokens.

Together, these results demonstrate that MIFO not only surpasses existing baselines in performance but also produces models that are more efficient in both data usage and output length, offering stronger reasoning capability and greater efficiency for training and deployment.

## 5.3 ABLATION STUDY

Table 3 reports an ablation on Qwen2.5-1.5B-Math model. We start from the base model and progressively add three components: interleaved SFT with RL (Interleave), entropy-based token selection (ES), and parameter freezing (PF) to mitigate SFT–RL interference. **+ Interleave.** Simply

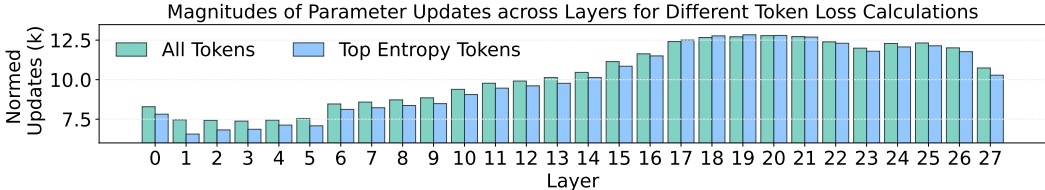

Figure 7: High-entropy token selection mitigates the magnitude of SFT updates for 1.5B model.

interleaving SFT with RL already boosts Overall accuracy from 15.0 to 34.0, though at the cost of longer outputs (4421 tokens). **+ Interleave + ES.** Adding entropy selection improves to 39.2 overall and shortens responses to 3246 tokens, confirming that focusing on high-uncertainty tokens reduces redundancy. **+ Interleave + PF.** Parameter freezing yields stable gains (Overall 38.4) and the best AIME-25 score (14.5), but still produces long traces (4490 tokens). **MIFO.** Combining ES and PF achieves the best Overall accuracy **40.0** and the shortest outputs (2518). It delivers top scores on multiple test sets, showing that ES and PF are complementary in improving reasoning performance.

## 5.4 ANALYZING

**Templates Ablation.** Besides using the LUFFY template (Yan et al., 2025) for results in Section 5.2, we evaluate MIFO under different reasoning templates to assess robustness in real-world deployments. Figure 6 reports performance for MIFO and the strongest baselines with no template (left) and with the Qwen system template (middle). MIFO's performance drops about 10 points when using simple templates, yet it still surpasses the baselines with best response efficiency. Detailed results by dataset and results with the Prime template are provided in Appendix E.2.

**MIFO Mitigates the Forgetting.** MIFO is motivated by the observation that SFT produces redundant parameter updates, whereas RL induces more parsimonious updates. We therefore reduce the SFT data size and limit updates by freezing parameters. Both components explicitly limit SFT updates, while whether computing the SFT loss only on high-entropy tokens decreases updates is uncertain. To quantify the effect, we analyze parameter changes under pure SFT with two objectives: (i) loss on full-token and (ii) loss on high-entropy–only, and we plot layer-wise changes. Figure 7 shows that for most layers the high-entropy objective yields noticeably smaller weight updates than the full-token objective. Only layers 17–19 exhibit slightly larger changes under the high-entropy objective. Overall, high-entropy token selection mitigates SFT updates on magnitude.

**Parameter Freezing Shortens Response.** Mirroring Section 5.3, we conduct an ablation to identify which components improve response efficiency and present results in Figure 8. Comparing **Interleave** with **Interleave+ES**, we observe that adding ES increases rollout length. We hypothesize that emphasizing high-entropy tokens during SFT encourages exploration of alternative reasoning branches (Wang et al., 2025a), which expands the trace. Comparing **Interleave+PF** with **Interleave**, PF yields shorter responses in most training stages, especially in the final epoch (three in total). This suggests that reducing SFT updates also improves efficiency, since SFT data are typically longer due to being sourced from stronger models with longer CoT. Finally, comparing **MIFO** with **Interleave+ES**, we find that PF substantially reduces response length, demonstrating its effectiveness for efficiency.

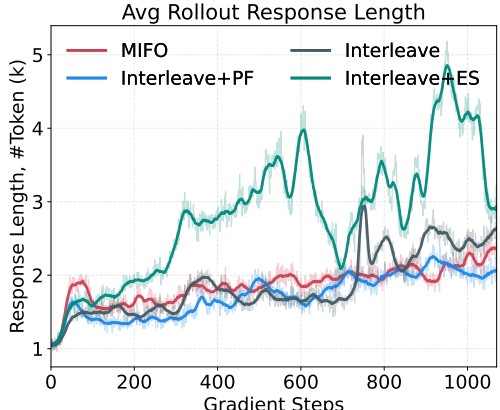

Figure 8: Rollout response length at training.

In addition to the above experiments, we further provide the experimental results on hyperparameter analysis, training records for dynamic analysis in Appendix E, and discussions in Appendix F.

## 6 CONCLUSION

We identify a key risk in combining SFT with RL for reasoning post-training: SFT applies much larger parameter updates that overwrite knowledge acquired by RL, leading to forgetting. We also observe a complementary pattern that informs a remedy: SFT tends to produce redundant updates, whereas RL induces more salient changes. Guided by these insights, we propose **MIFO**, a plug-and-play framework with two components: data processing and parameter freezing. MIFO is algorithm-agnostic, achieves state-of-the-art reasoning with reasoning efficiency, and requires significantly less training data compared to the previous SOTA method.

## ETHICS STATEMENTS

Our work aims to improve the reasoning capability of LLMs on mathematical problems, which pose no safety risks. The study does not involve collecting sensitive data. All experiments use public training corpora and standard math reasoning benchmarks (AIME-24, AIME-25, AMC, Olympiad, MATH-500, MMLU-Pro).

## REPRODUCIBILITY STATEMENT

Implementation details are provided in Section 5.1 and Appendix D to facilitate reproducibility within the research community. We also provide the log information of the training dynamics and hyperparameter analysis in Appendix E for reproducibility reference.

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

## A  THE USE OF LARGE LANGUAGE MODELS (LLMS)

We strictly adhere to the ICLR Code of Ethics and only utilize the Large Language Model to polish the paper for grammar and expression checks.

## B  LIMITATION AND FUTURE WORKS

Our study focuses on small (1.5B) and medium (7B) models due to limited computational resources. Scaling the method to larger models is a promising direction for future work. Our approach primarily targets full-parameter SFT; when applying parameter-efficient fine-tuning (PEFT), the freezing strategy may need to be adjusted to prevent catastrophic forgetting between SFT and RL. As stronger SFT and RL algorithms continue to emerge, we release this plug-and-play framework to encourage the community to explore improved post-training reasoning with advanced algorithm combinations.

## C  THEORETICAL INSIGHTS AND PROOF

Here we provide theoretical insights trying to explain why SFT exhibits greater redundancy in parameter updates than RL. The following discussion and proofs rest on simplifying assumptions intended to interpret our empirical findings, instead of providing a rigorous theoretical guarantee. Because large language models involve extremely high-dimensional parameters and massive training data, fully rigorous modeling is challenging.

### C.1  REDUNDANCY RATIO DEFINITION

In this section, we introduce Lemma 1 to describe the *decision–redundancy ratio* (DR) as how much a parameter update exceeds the minimal necessary change to adjust the decision margin of logits for the right token prediction. For a given input and target token, we look at the strongest non-target competitor under the current parameters and measure the logit margin between the target and that competitor. Assuming the margin changes approximately linearly with small parameter moves and the same competitor remains active, there exists a unique, shortest move in parameter space that achieves any desired margin increase; this move points along the margin's gradient. DR is then defined as the size of the actual parameter change divided by the size of the shortest move. A larger DR means the training trajectory moved farther than necessary to achieve the same decision margin, indicating more redundant updating.

**Lemma 1.** *Fix a context $x$ and target token $y \in \{1, \ldots, V\}$ with vocabulary size $V$. Let $\theta \in \mathbb{R}^d$ be model parameters and $\theta_0$ a reference point (e.g., pre-update). For logits $z_\theta(x) \in \mathbb{R}^V$ and probabilities $p_\theta = \mathrm{softmax}(z_\theta)$, define the active competitor at $\theta_0$ as $k^\star = \arg\max_{k \neq y} z_{\theta_0, k}(x)$. The (logit) decision margin is $m(\theta) = z_{\theta, y}(x) - z_{\theta, k^\star}(x)$, with $m_0 = m(\theta_0)$, and its parameter gradient is $g = \nabla_\theta m(\theta_0) \in \mathbb{R}^d$. Assume (A1) local linearity $m(\theta_0 + \Delta) \approx m_0 + g^\top \Delta$ for small $\Delta \in \mathbb{R}^d$ and (A2) competitor stability (the same $k^\star$ remains active locally). Then for any desired margin $\varepsilon > 0$, the minimal parameter displacement $\Delta^\star$ that achieves margin $\varepsilon$ is*

$$\Delta^\star = \arg\min_\Delta \{\|\Delta\|_2 : g^\top \Delta \geq \varepsilon - m_0\} = \frac{\varepsilon - m_0}{\|g\|_2^2} g, \qquad \|\Delta^\star\|_2 = \frac{\varepsilon - m_0}{\|g\|_2},$$

*and for any training trajectory with net update $\Delta\theta = \theta_T - \theta_0$, we have decision–redundancy ratio*

$$\mathrm{DR}_x \equiv \frac{\|\Delta\theta\|_2}{\|\Delta^\star\|_2} = \|\Delta\theta\|_2 \cdot \frac{\|g\|_2}{\varepsilon - m_0}. \tag{1}$$

*Proof.* Let $\beta := \varepsilon - m_0$. Under (A1) local linearity (a first-order Taylor approximation, which is standard for analyzing the local geometry of optimization landscapes (Bottou et al., 2018).) and (A2) competitor stability, achieving the (linearized) target margin $\varepsilon$ is equivalent to requiring

$$g^\top \Delta \geq \beta,$$

where $g = \nabla_\theta m(\theta_0)$. Hence the minimal-displacement problem is

$$\Delta^\star = \min_{\Delta \in \mathbb{R}^d} \|\Delta\|_2 \quad \text{s.t.} \quad g^\top \Delta \geq \beta.$$

If $\beta \leq 0$, then $\Delta^\star = 0$ is feasible and optimal (the target margin is already met), and the result is trivial. In the remainder, assume $\beta > 0$ and $g \neq 0$.

For any feasible $\Delta$, by Cauchy–Schwarz,

$$\beta \ \leq \ g^\top \Delta \ \leq \ \|g\|_2 \, \|\Delta\|_2 \quad \Longrightarrow \quad \|\Delta\|_2 \ \geq \ \frac{\beta}{\|g\|_2}.$$

Equality holds if and only if $\Delta$ is colinear and aligned with $g$, i.e., $\Delta = \alpha g$ with $\alpha \geq 0$. Choosing

$$\alpha \ = \ \frac{\beta}{\|g\|_2^2} \quad \Rightarrow \quad \Delta^\star \ = \ \frac{\beta}{\|g\|_2^2} \, g, \qquad \|\Delta^\star\|_2 \ = \ \frac{\beta}{\|g\|_2}.$$

Thus $\Delta^\star$ achieves the lower bound and is the unique minimizer (any deviation from the $g$ direction increases the norm for fixed inner product $g^\top \Delta = \beta$).

Now consider any training trajectory with net update $\Delta\theta = \theta_T - \theta_0$ that attains the target margin in the linearized model, so $g^\top \Delta\theta \geq \beta$. By optimality of $\Delta^\star$, $\|\Delta\theta\|_2 \geq \|\Delta^\star\|_2$. Therefore, for $\beta > 0$,

$$\mathrm{DR}_x \ \equiv \ \frac{\|\Delta\theta\|_2}{\|\Delta^\star\|_2} \ = \ \|\Delta\theta\|_2 \cdot \frac{\|g\|_2}{\varepsilon - m_0}.$$

(When $\beta \leq 0$, we have $\Delta^\star = 0$ and the margin is already satisfied.) $\qquad\square$

## C.2 REDUNDANCY ANALYSIS OF SFT AND RL

Based on the decision–redundancy ratio (DR) defined in the previous section, we now analyze the properties of SFT and RL based on their gradient. We first borrow the policy gradient form of SFT from Wu et al. (2025), and then compare their decision–redundancy ratios.

**SFT Gradient.** Let $\mathcal{D} = \{(x, y^*)\}$ denote a corpus of annotated demonstrations, where $y^*$ is the reference response for query $x$. SFT minimizes the cross-entropy loss:

$$\mathcal{L}_{\mathrm{SFT}}(\theta) = \mathbb{E}_{(x,y^*)\sim\mathcal{D}}[-\log \pi_\theta(y^* \mid x)],$$

and its gradient is:

$$\nabla_\theta \mathcal{L}_{\mathrm{SFT}}(\theta) = \mathbb{E}_{(x,y^*)\sim\mathcal{D}}[-\nabla_\theta \log \pi_\theta(y^* \mid x)]. \tag{2}$$

**RL Gradient.** Let $y$ denote a rollout response sampled from the policy model $\pi_\theta(\cdot \mid x)$ for query $x$. Given a reward function $r(x, y) \in \mathbb{R}$, the policy objective is

$$J(\theta) = \mathbb{E}_{x\sim\mathcal{D}_x,\, y\sim\pi_\theta(\cdot|x)}[r(x, y)],$$

and its policy gradient is

$$\nabla_\theta J(\theta) = \mathbb{E}_{x\sim\mathcal{D}_x,\, y\sim\pi_\theta(\cdot|x)}[\nabla_\theta \log \pi_\theta(y \mid x)\, r(x, y)]. \tag{3}$$

**Rewriting SFT Gradient as Policy Gradient.** Here we directly quote the conclusion from Wu et al. (2025) to rewrite SFT gradient to policy gradient. The SFT gradient in Equation 2 taken under the fixed demonstration distribution is convert it to an on-policy expectation by inserting an importance weight that compares the expert (Dirac Delta) distribution with the model distribution:

$$\mathbb{E}_{(x,y^*)\sim\mathcal{D}}[-\nabla_\theta \log \pi_\theta(y^* \mid x)] = \mathbb{E}_{x\sim\mathcal{D}_x} \underbrace{\mathbb{E}_{y\sim\pi_\theta(\cdot|x)}\left[\frac{\mathbb{I}[y = y^*]}{\pi_\theta(y \mid x)}\left(-\nabla_\theta \log \pi_\theta(y \mid x)\right)\right]}_{\text{resample + reweight}}. \tag{4}$$

Define the auxiliary variables

$$w(y \mid x) = \frac{1}{\pi_\theta(y \mid x)}, \qquad r(x, y) = \mathbb{I}[y = y^*].$$

Reorganize Equation 4 and rewrite it using the above auxiliary variables; we obtain

$$\nabla_\theta \mathcal{L}_{\mathrm{SFT}}(\theta) = -\mathbb{E}_{x\sim\mathcal{D}_x,\, y\sim\pi_\theta(\cdot|x)}\left[w(y \mid x)\, \nabla_\theta \log \pi_\theta(y \mid x)\, r(x, y)\right]. \tag{5}$$

This form of the SFT gradient now closely aligns with the policy gradient in Equation 3: Conventional SFT can be viewed as an on-policy gradient where the reward is an indicator of matching the expert response, modulated by an importance weight $1/\pi_\theta$.

Then, we will utilize obtained conclusions to analyze DR of SFT and RL.

**Theorem 1.** *Define the reward function as $r$ for the policy model $\pi_\theta$. For a input context $x$, policy model output token $y_t$ according to $x$ and target token $y_t^*$, if $\mathbb{I}[y_t = y_t^*] \geq r\pi_\theta(y_t \mid x)$, SFT (policy model expression) and RL predict $y^*$ that both attain the same target margin $\varepsilon$ satisfy*

$$\mathrm{DR}_x^{\mathrm{SFT}} \geq \mathrm{DR}_x^{\mathrm{RL}}.$$

*Proof.* According to Equation 1 in Lemma 1, we have the following DR for SFT and RL:

$$\mathrm{DR}_x^{SFT} \equiv \frac{\|\Delta\theta^{SFT}\|_2}{\|\Delta^\star\|_2} = \|\Delta\theta^{SFT}\|_2 \cdot \frac{\|g\|_2}{\varepsilon - m_0} = \|\eta\nabla_\theta\mathcal{L}_{\mathrm{SFT}}(\theta)\|_2,$$

$$\mathrm{DR}_x^{RL} \equiv \frac{\|\Delta\theta^{RL}\|_2}{\|\Delta^\star\|_2} = \|\Delta\theta^{RL}\|_2 \cdot \frac{\|g\|_2}{\varepsilon - m_0} = \|\eta\nabla_\theta J(\theta)\|_2.$$

Then we utilize the sentence level policy gradient expressions in Equation 4 and Equation 3. Here we make slight change from the sentence level expression to token level expression here to simplify the illustration and have:

$$\frac{\mathrm{DR}_x^{SFT}}{\mathrm{DR}_x^{RL}} = \frac{\|\nabla_\theta\mathcal{L}_{\mathrm{SFT}}\|_2}{\|\nabla_\theta J(\theta)\|_2} = \frac{\mathbb{I}[y_t = y_t^*]}{r\pi_\theta(y_t \mid x)} \geq 1 \tag{6}$$

So we have

$$\mathrm{DR}_x^{\mathrm{SFT}} \geq \mathrm{DR}_x^{\mathrm{RL}}.$$

$\square$

Theorem 1 provides the condition when $\mathrm{DR}_x^{SFT}$ will larger then $\mathrm{DR}_x^{RL}$. Since when SFT updates, the gradient in Equation 4 can't be zero, therefore $\mathbb{I}[y_t = y_t^*] = 1$. And for hard reasoning case, the policy probability $\pi_\theta(y_t \mid x) \ll 1$ while reward function $r$ is usually bounded (PPO/GRPO), therefore $\mathbb{I}[y_t = y_t^*] \geq r\pi_\theta(y_t \mid x)$, and SFT has more updating redundancy compared to RL.

Our theoretical analysis, formalized in Lemma 1 and Theorem 1, provides a causal mechanism for the empirical phenomena observed in Section 3.1. The core theoretical finding is that the Decision-Redundancy (DR) on logits for predicting the target token of the SFT is intrinsically higher than that of the RL after parameter updates (i.e., $\mathrm{DR}^{\mathrm{SFT}} \geq \mathrm{DR}^{\mathrm{RL}}$). This conclusion directly explains why SFT exhibits *sparsification robustness*: Theorem 1 predicts SFT has a high DR, implying its parameter updates are "redundant." This redundancy allows SFT to maintain performance even when 50% of updates are pruned (as shown in Figures 1 and Figure 2), as it has sufficient redundant updates to secure the correct token prediction. Conversely, RL's low DR suggests its updates are *parsimonious* and efficient, which explains its fragility and immediate performance degradation under similar pruning conditions. Since calculating the exact DR is computationally intractable for large language models, our sparsification experiments serve as the empirical proxy to validate this theory. Theorem 1 predicts this imbalance, and the experiments (Figures 1 and Figure 2) confirm it by demonstrating SFT's better tolerance to pruning.

## D    IMPLEMENTATION DETAILS

**Method Implementation**    We implement RL and SFT following widely used settings. For RL, we remove the KL penalty, length normalization, and standard-error normalization, as in Dr.GRPO (Liu et al., 2025c) and LUFFY (Yan et al., 2025). GRPO hyperparameters are: rollout batch size 128, number of rollouts per query 8, update batch size 64, learning rate $1 \times 10^{-6}$, rollout temperature 1.0, and entropy coefficient 0.001. For SFT, the training batch size is 128 and the learning rate is $1 \times 10^{-5}$. For MIFO, the rollout-accuracy threshold for buffering is $p = \frac{1}{8}$, the high-entropy token ratio is $\rho = 0.2$, the TopK freezing fraction is $k = 0.5$, and the decay coefficient is $\alpha = 0.5$. The buffer size is set to be 64, with a study on buffer size showing in Table 5. Pseudocode of MIFO is provided in Algorithm 1.

---

**Algorithm 1 MIFO**: MItigating FOrgetting between SFT and RL

---

**Require:** Policy $\pi_\theta$, old policy $\pi_{\theta_{\text{old}}}$, reward $R$, entropy ratio $\rho$, decay $\alpha$, Buffer size $S$, SFT data D.

    Initialize $\text{Buffer}_{\text{FT}} \leftarrow \emptyset$, $C_0^j = 0$ for all $j$, Define SFT $\rightarrow$ RL iteration $i$ as $\text{Interval}_i^{RL \rightarrow SFT}$

    **for** each $\text{Interval}_i^{RL \rightarrow SFT}$ interval iteration $i$ **do**

        Record $\theta_{RL,start,i}^{\text{j}}$

        **for** each question $q$ **do**

            $\pi_{\theta_{\text{old}}}(q)$ rollout, compute $R(o_i)$ and advantages $A_i$. Update $\pi_\theta$ via GRPO

            **if** $\text{acc}(q) < a$ **and** $\text{extract}(D(q)) = a$ **then**

                Add $(q, D(q))$ to $\text{Buffer}_{\text{FT}}$

            **end if**

        **end for**

        Record $\boldsymbol{\theta}_{RL,end,i}^{\text{j}}$

        $\Delta\boldsymbol{\theta}_{\text{RL},i}^{j} \leftarrow \left\|\theta_{\text{RL,end},i}^{j} - \theta_{\text{RL,start},i}^{j}\right\|_2$, $\tilde{\boldsymbol{C}}_i = \alpha\,\boldsymbol{C}_{i-1} + (1-\alpha)\,\Delta\boldsymbol{\theta}_{\text{RL},i}$

        Obtain the mask $\mathbf{M}_i^j$: $\mathbf{M}_i^j = \mathbf{1}[\,j \in \mathcal{I}_i\,]$; $\mathbf{M}_i^j = \mathbf{1}[\,j \in \mathcal{I}_i\,]$, $\mathbf{C}_i = \mathbf{M}_i \odot \tilde{\mathbf{C}}_i$,

        Freeze Selected parameters $\nabla_{\boldsymbol{\theta}_i}\mathcal{L}_{\text{SFT}} \leftarrow (\mathbf{1} - \mathbf{M}_i) \odot \nabla_{\boldsymbol{\theta}_i}\mathcal{L}_{\text{SFT}}$.

        SFT only calculate loss for tokens with $H_t \geq \tau_\rho^q$ in $\text{Buffer}_{\text{FT}}$, and unfreeze.

    **end for**

---

Table 4: GPU hours for different methods.

| Method | SFT | RL | SFT-RL | LUFFY | LUFFY+ | ReLIFT | MIFO |
|---|---|---|---|---|---|---|---|
| GPU Hour | 8×8 | 40×8 | 67×8 | 77×8 | 130×8 | 52×8 | 74×8 |

**Training** Following prior work (Yan et al., 2025; Ma et al., 2025; Fu et al., 2025; Cui et al., 2025; Liu et al., 2025c; Hu et al., 2025), we train on Qwen2.5-Math (1.5B and 7B) (Yang et al., 2024). As in (Fu et al., 2025; Yan et al., 2025; Ma et al., 2025), we increase the RoPE base from 10,000 to 40,000 and extend the context window to 16,384 tokens to encourage exploration. Models are trained for three epochs on $4 \times 8$ A100 GPUs using the VERL framework (Sheng et al., 2025). The reward is binary:

$$R = \begin{cases} 1, & \text{if the answer is correct,} \\ 0, & \text{otherwise.} \end{cases}$$

The training GPU hours comparison information is provided in Table 4. Note that the hours information is based on 8 GPUs to fairly compare with baselines.

**Evaluation** Except for Oat-Zero and OpenReasoner-7B, we independently evaluate publicly released baseline checkpoints to reproduce and verify prior results. The remaining baseline scores are taken from (Ma et al., 2025), which adopts the same evaluation protocol as MIFO. All evaluations are conducted with vLLM (Sheng et al., 2025), using a temperature of 0.6 and a maximum sequence length of 8,192 tokens. Additional dataset statistics are provided in Table 6.

**Chat Template** For our main experiments, we follow (Ma et al., 2025; Yan et al., 2025; Fu et al., 2025) and adopt the LUFFY chat template across all training paradigms. This unified system prompt encourages systematic reasoning—analyze, summarize, explore, reassess, and refine—as illustrated in Figure 10. We also evaluate with other popular templates shown in Figure 10 to assess robustness under template shifts.

Table 5: The average pass@1 accuracy for training Qwen2.5-1.5B-Math with 1 epoch.

| Buffer Size | 8 | 16 | 64 | 128 | 256 |
|---|---|---|---|---|---|
| Avg | 32.4 | 34.4 | **36.3** | 36.1 | 35.8 |

Table 6: The statistics of evaluation datasets

| Dataset | #Test | Task Type | Domain | #k |
|---------|-------|-----------|--------|-----|
| AIME24 | 30 | Math competition | Mathematics | 32 |
| AIME25 | 30 | Math competition | Mathematics | 32 |
| AMC | 83 | Math competition | Mathematics | 32 |
| MATH-500 | 500 | Mathematical reasoning | Mathematics | 1 |
| Olympiad | 674 | Math competition | Mathematics | 1 |
| MMLU-Pro | 12,102 | Multi-task understanding | Multidisciplinary | 1 |

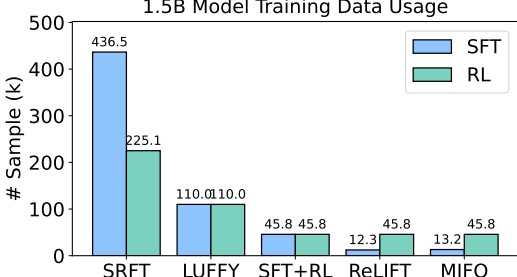

Figure 9: Data usage for training Qwen2.5-1.5B-Math model

**Data Usage for Training** We report training data usage for the 7B model in Figure 6 (right), and we provide additional details for the 1.5B model in Figure 9. Because the 1.5B model has weaker reasoning capability, its lower rollout accuracy leads to a slight increase in SFT data consumption. Baseline data usage is compiled from the corresponding papers and from dataset statistics on Hugging Face. For the LUFFY baseline, we use the LUFFY[†] variant (Yan et al., 2025), which has stronger reasoning performance, to reproduce results and record the associated data usage.

# E ADDITIONAL EXPERIMENTS

## E.1 HYPERPARAMETER ANALYSIS

Our method uses four hyperparameters: the rollout-accuracy threshold $p$ for buffering questions, the high-entropy token ratio $\rho$, the TopK freezing fraction $k$, and the history coefficient $\alpha$. As shown in Figures 11, these hyperparameters are insensitive and easy to tune. In Figure 11 (left), the model performs best at $p = \frac{2}{8}$. Since performance is comparable across nearby settings, we choose $p = \frac{1}{8}$ to reduce SFT data usage. Since the limit of computational resources, this analysis is conducted for only 1 epoch of training.

## E.2 TEMPLATE ABLATION STUDY

In real-world deployment, users may not provide carefully designed chat prompts for LLM reasoning. It is therefore important to assess robustness in the wild. We evaluate MIFO under multiple reasoning templates to test whether it maintains strong performance without curated prompts. In Section 5.4, we reported average reasoning accuracy and response length under the *No template* and *Qwen* (Yang et al., 2024) settings. Here we add results with another popular template, *Prime* (Cui et al., 2025), which is commonly used for RL-trained reasoning models, and we provide per-dataset scores in Table 7. Table 7 further indicates that our method is template-robust, outperforming competitive baselines across all three templates.

## E.3 TRAINING DYNAMICS

**Batch Solve** Figure 12 presents training dynamics for ablated models on *batch solved rate* (left) and *batch all solved rate* (right). For both **batch solved rate** and **batch all solved rate**, Interleave

> **Chat Template (Qwen)**
>
> Please reason step by step, and put your final answer within \\box{}.
> **Question:** {question}
> **Answer:** {answer}

> **Chat Template (Prime)**
>
> A conversation between User and Assistant. The user asks a question, and the Assistant solves it. The assistant first thinks about the reasoning process in the mind and then provides the user with the answer. The reasoning process and answer are enclosed within <think> </think> and <answer> </answer> tags, respectively, i.e., <think> reasoning process here </think> <answer> answer here </answer>
> **Question:** {question}
> **Answer:** {answer}

> **Chat Template (LUFFY)**
>
> Your task is to follow a systematic, thorough reasoning process before providing the final solution. This involves analyzing, summarizing, exploring, reassessing, and refining your thought process through multiple iterations. Structure your response into two sections: Thought and Solution. In the Thought section, present your reasoning using the format:"\n {thoughts} \n". Each thought should include detailed analysis, brainstorming, verification, and refinement of ideas. After "\n" in the Solution section, provide the final, logical, and accurate answer, clearly derived from the exploration in the Thought section. If applicable, include the Answer in \boxed{} for closed-form results like multiple choices or mathematical solutions.
> **Question:** {question}
> **Answer:** {answer}

Figure 10: Chat template details.

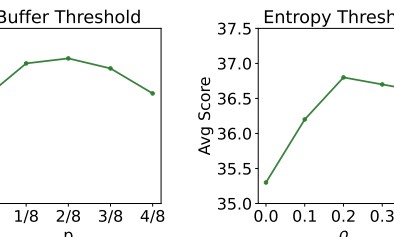 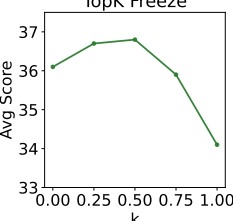 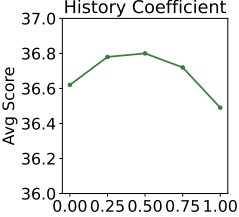

Figure 11: Hyperparameter analysis.

+ ES has comparable performance with PF, and they all outperform Interleave. **MIFO** outperforms other ablated models at most of the steps, indicating the effectiveness of combining each module: increasing the rollout accuracy during the RL training.

**Reward Value**    Figure 13 (left) reports the average rollout reward during training. All variants improve rapidly at the start, then separate modestly mid-training. **Interleave+ES** and **Interleave+PF** have comparable reward value, outperforming **Interleave** and falling behind MIFO. The combined (**MIFO**) stays at or near the top throughout and converges to the highest point. These trends indicate that each module contributes positively to MIFO.

**Average Validation Score**    Similarly, Figure 13 (right) supplements the ablation in Section 5.3. The ablated models **Interleave+ES** and **Interleave+PF** generally surpass the **Interleave** baseline, confirming the effectiveness of ES and PF. The combined **MIFO** outperforms the ablations for most

Table 7: Performance comparison on Qwen-Math-2.5 with different templates. **Bold** and underline indicate the best and second-best results, respectively.

| Templates | Model | AIME-24 | AIME-25 | AMC | MATH-500 | Olympiad | MMLU-Pro | Average |
|---|---|---|---|---|---|---|---|---|
| No | ReLIFT | 23.3 | 14.1 | 54.2 | 75.0 | 41.8 | 16.4 | 37.6 |
| | LUFFY | 25.4 | 14.1 | 55.9 | 75.4 | 41.0 | 16.3 | 38.0 |
| | SRFT | **27.7** | 14.3 | **59.2** | 77.8 | 42.9 | 21.7 | 40.6 |
| | MIFO | 26.1 | **17.7** | 58.7 | **79.6** | **43.4** | **22.9** | **41.4** |
| Qwen | ReLIFT | 25.3 | 14.7 | 59.5 | 76.6 | 40.7 | 45.1 | 43.7 |
| | LUFFY | 25.3 | 13.0 | 56.6 | 76.4 | 40.2 | 44.8 | 42.7 |
| | SRFT | 27.3 | 15.2 | 58.9 | 77.0 | 42.1 | 45.3 | 44.3 |
| | MIFO | **28.2** | 16.3 | **61.9** | **77.4** | **42.4** | **45.7** | **45.3** |
| Prime | ReLIFT | 27.2 | **21.9** | 63.4 | 86.0 | **52.1** | 49.2 | 50.0 |
| | LUFFY | 27.4 | 17.0 | 61.2 | 84.4 | 46.1 | 47.8 | 47.3 |
| | SRFT | 26.7 | 20.1 | 68.1 | 86.2 | 49.3 | 49.9 | 50.1 |
| | MIFO | **28.5** | **21.9** | **68.3** | **87.4** | 50.2 | **50.0** | **51.1** |

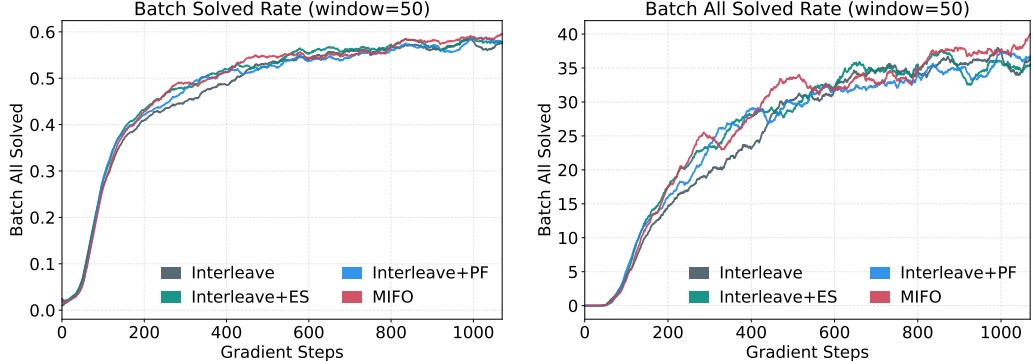

Figure 12: Training dynamics for ablated models on batch solving with running average.

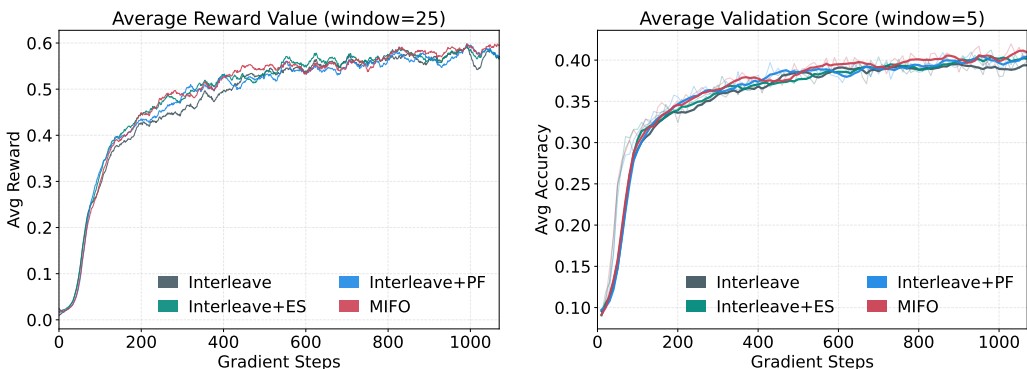

Figure 13: Training dynamics for ablated models on average reward value (left) and average validation scores (right) with running average.

of the training, supporting the choice to combine Interleave, ES, and PF for stronger reasoning performance.

### E.4 EXTRA UPDATING DROPPING EXPERIMENT

In Section 3, we demonstrated SFT redundancy using experiments that randomly drop parameter updates. Building on this insight, Section 4 introduces parameter freezing based on RL-derived parameter importance. While the initial experiment supports the redundancy claim, it leaves a gap between *random freezing* and *selective freezing*. To bridge this, we record cumulative RL-induced parameter changes and select the top-$K$ indices for dropping, using the same settings as in Section 3. Figure 14 shows no notable performance difference between random dropping and RL-based selective dropping. Thus, the redundancy conclusion remains valid for our design. We also observe that RL top-$K$ selection is slightly outperformed by random selection, suggesting partial overlap between parameters important for SFT and those important for RL.

The previous results were for a model trained for one epoch. Here, we further present post-hoc update sparsification results in Figure 15 for a model trained with three epochs to convergence at step 1070 (as Figure 13 confirms). Figure 15 indicates that SFT has more redundancy than RL, which aligns with our previous conclusion.

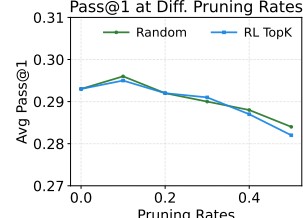

Figure 14: Test pass@1 after 1 epoch training.

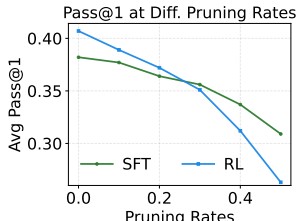

Figure 15: Test pass@1 after 3 epochs training.

### E.5 QUANTITATIVELY MEASURE THE FORGETTING

To quantitatively measure the forgetting for our claims in Section 3.2, we design an experiment to fairly measure the forgetting condition for different baselines and MIFO. We first train the model with RL for 50 steps (average test score=24.3), and then do training for multiple methods containing SFT objectives for 10 steps for evaluation. Figure 16 shows the final test accuracy after training: if the training objective has SFT (SFT, SRFT, LUFFY, ReLIFT) without forgetting handling, the model performances largely drop after the previous RL training, indicating the catastrophic forgetting caused by SFT, while MIFO experiences a performance gain after training, indicating its effectness on mitigating forgetting. Also, our ablation study in Section 5.3 provides another straight evidence on "mitigates forgetting". Comparing +Interleave and +Interleave+PF, we find that just freezing a large amount of parameters for SFT training not only avoid performance drop, but also has external markable improvements (+4.4). If our claim on mitigating forgetting is wrong, the performance wouldn't gain a large improvement after parameter freezing.

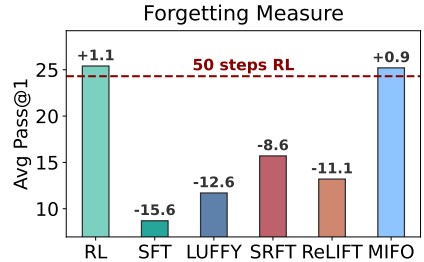

Figure 16: Average test pass@1 for measuring forgetting quantitatively.

### E.6 EXPERIMENTS ON ADDITIONAL MODEL, ALGORITHMS AND DOMAINS

We clarify that our description of MIFO as "algorithm-agnostic" refers to its structural compatibility as a plug-and-play method. MIFO treats the RL and SFT as modular components that provide two generic outputs: rollout accuracy and entropy for data selection, and parameter update magnitudes for the freezing mechanism. Since these two signals are inherent to any gradient-based RL or SFT method, the MIFO framework can mechanically integrate with them without modification to its core logic. Therefore, the validity of our proposed mechanism, which mitigates catastrophic forgetting by freezing parameters that changed significantly during the RL phase, holds regardless of the specific optimizer used to generate those changes. We utilized GRPO as the primary instantiation because it is a popular method for reasoning post-training. Furthermore, we provide additional experiments on

Table 8: Performance comparison on Qwen2.5-7B-Insturct-1M trained on coding task. **Bold** and underline indicate the best and second-best results, respectively.

| Model | LCB | HumanEval+ | MBPP+ | Avg |
|---|---|---|---|---|
| Base | 24.0 | 80.5 | 66.7 | 57.1 |
| SFT | 24.5 | 81.7 | 66.9 | 57.7 |
| RL | 28.6 | 84.8 | 70.1 | 61.2 |
| SFT→RL | 28.8 | 83.5 | 71.4 | 61.2 |
| MIFO | **29.1** | **86.6** | **73.0** | **62.9** |

Table 9: Performance comparison on Qwen2.5-Math-7B for scientific/knowledge reasoning tasks. **Bold** and underline indicate the best and second-best results, respectively.

| | ARC-c | GPQA | MMLU-Pro | Avg |
|---|---|---|---|---|
| Base | 70.3 | 24.7 | 34.1 | 43.0 |
| SFT | 75.2 | 24.7 | 42.7 | 47.5 |
| RL | 83.2 | 40.4 | 49.3 | 57.6 |
| SFT→RL | 72.4 | 24.2 | 37.7 | 44.8 |
| LUFFY | 80.5 | 39.9 | 52.6 | 57.7 |
| SRFT | **85.3** | 46.4 | 54.1 | **61.9** |
| MIFO | 83.8 | **47.4** | **54.3** | 61.8 |

Table 10: Performance comparison on Llama3.2-8B on math reasoning tasks. **Bold** and underline indicate the best and second-best results, respectively.

| | AIME24 | AIME25 | AMC | MATH | Olympiad | MMLU-Pro | Avg |
|---|---|---|---|---|---|---|---|
| SFT | 0.8 | 1.5 | 11.5 | 28.6 | 8.6 | 28.2 | 13.2 |
| RL | 1.8 | 0.0 | 10.8 | 28.2 | 7.3 | 39.4 | 14.6 |
| ReLIFT | 1.3 | 0.2 | 11.9 | 35.2 | 11.0 | 44.2 | 17.3 |
| MIFO | **2.1** | **2.5** | **13.9** | **40.6** | **12.3** | **46.6** | **19.7** |

multiple SFT and RL combinations (Yu et al., 2025; Wu et al., 2025; Guo et al., 2025) in Table 11, which show the effectiveness of MIFO in improving different SFT-RL combinations.

We present additional experiments to demonstrate MIFO's effectiveness in combining SFT and RL training across diverse models and domains. This is confirmed by the scientific/knowledge-based reasoning results (Table 9), Llama 3.2 8B performance (Table 10, following the setup in Section 5.1) including template sensitivity analysis (Figure 17), and Qwen2.5-7B-Instruct-1M coding results (Table 8). Following Fu et al. (2025); Yan et al. (2025), we utilize scientific/knowledge-based reasoning tasks as an out-of-distribution (OOD) evaluation. For coding, we employ the Code-R1 pipeline (Liu & Zhang, 2025) to evaluate performance on LCB (Jain et al., 2024), HumanEval+, and MBPP+ (Liu et al., 2023). Quantitatively, Table 8 demonstrates MIFO's superior pass@1

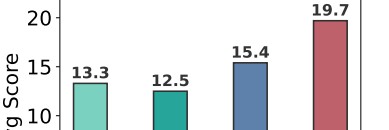

Performance w. Different Templates

Figure 17: Average test pass@1 for Llama 3.2-8B w. different templates.

performance on code generation tasks using Qwen2.5-7B-Instruct-1M. MIFO achieves remarkable performance across all benchmarks, reaching an average score of 62.9%, which is a 1.7% improvement over the SFT→RL baseline. Table 10 highlights consistent improvements in math reasoning using the Llama 3.2 8B model, where MIFO outperforms the competitive SFT+RL baseline, ReLIFT, by 2.4 points on average. Finally, Table 9 confirms MIFO's effectiveness on scientific and knowledge-based benchmarks, showing a substantial improvement of 17 points on average over the SFT→RL baseline. Furthermore, MIFO performs comparably to SRFT on the average score while requiring significantly less data, as detailed in Section 5.2. Collectively, these extensive experiments validate MIFO as a versatile solution for the SFT-RL forgetting problem, demonstrating robust generalization and consistent gains across varying model sizes, architectures, and complex reasoning domains.

# F  DISCUSSION

## F.1  EXPERIMENT DESIGN FOR OBSERVATIONS

Here we further justify the experimental design in Section 3.1. Using both *online* and *post-hoc* sparsification probes complementary forms of redundancy and yields a stronger diagnosis of SFT

Table 11: Performance comparison on Qwen2.5-Math-3B with different SFT and RL method combinations. **Bold** and underline indicate the best and second-best results, respectively.

| Models | SFT+GRPO | DFT+GRPO | SFT+DAPO | DFT+DAPO |
|--------|----------|----------|----------|----------|
| SFT | 23.6 | 23.1 | 23.6 | 23.1 |
| RL | 27.5 | 27.5 | 27.8 | 27.8 |
| SFT-RL | 27.9 | 27.3 | 28.2 | 27.9 |
| MIFO | **30.7** | **29.2** | **30.5** | **29.7** |

and RL learning. *Online sparsification* evaluates training-time resilience. If performance remains stable when randomly dropping half of the gradient coordinates, the learning signal and optimizer dynamics are overparameterized (gradient-level redundancy) rather than dependent on precise dense updates. *Post-hoc sparsification* evaluates endpoint redundancy. If the final predictor maintains accuracy after pruning the net parameter change, a substantial portion of the learned displacement $\Delta$ is superfluous (parameter-change redundancy). This separates trajectory effects from the equivalence class of solutions at convergence. Together, these two regimes provide orthogonal evidence for redundancy within each training paradigm.

### F.2 WHY FREEZING INSTEAD OF SFT VARIANTS

Prior work on fine-tuning LMs offers a key insight: *fully fine-tuning is not always optimal; selectively updating a subset of parameters can match or even surpass full-model tuning*. Evidence for layer heterogeneity suggests that focusing updates on important components is preferable to uniform tuning (Zhang et al., 2022; Lee et al., 2019). ULMFiT (Howard & Ruder, 2018) progressively unfreezes layers to retain prior knowledge and reduce catastrophic forgetting. Parameter-efficient methods (Houlsby et al., 2019; Li & Liang, 2021; Hu et al., 2022) train only a small fraction of parameters yet achieve comparable performance. BitFit (Ben Zaken et al., 2022) reaches strong results by updating only bias terms. AdapterDrop (Rücklé et al., 2021) shows that adapting only important layers can be more effective and efficient.

These lessons highlight inherent drawbacks of SFT, including overfitting and unequal utility across parameters. Our dropout-like selective freezing for SFT (Srivastava et al., 2014; Baldi & Sadowski, 2013) both protects RL-acquired knowledge and reduces overfitting risk (Section 3.1). While reducing data or lowering the learning rate can also lessen SFT-induced forgetting, such approaches risk underutilizing SFT knowledge.

### F.3 THE RELATIONSHIP AND DIFFERENCE WITH PREVIOUS WORKS

Our buffer is built on the buffer design of ReLIFT (Ma et al., 2025). ReLIFT performs online SFT using a buffer restricted to $acc(q) = 0$, i.e., questions $q$ that GRPO fails to solve. This choice is motivated by the empirical finding that SFT helps on unsolved questions but can lose its benefit once the policy already solves them. In our study, we find that easier questions still contain useful supervision. Under ReLIFT's setting, the gains from such questions may be offset by catastrophic forgetting, since ReLIFT uniformly learns all tokens, including high-confidence tokens in easy cases. In contrast, our approach sets a threshold $acc(q) \leq p$ and combines high-entropy token selection with parameter freezing. These modules mitigate SFT–RL forgetting, enlarge the usable data pool, and exploit more of the available reasoning signal. Compared with ReLIFT, our contribution is not merely introducing a new threshold. By targeting uncertain tokens and protecting RL-sensitive parameters, we unlock additional data utility and push beyond the prior ceiling on bufferable examples.

### F.4 RESPONSE LENGTH DISCREPANCY OF **MIFO** AND **MIFO**[†]

Table 1 and 2 show that **MIFO** produces shorter response length than **MIFO**[†] for the 7B model, while **MIFO**[†] yields shorter response length than **MIFO** for the 1.5B model. To interpret these results, we first note the baseline behaviors: SFT tends to be extremely verbose ($>$10k tokens) while RL encourages conciseness ($<$1k tokens). Therefore, the output length serves as a proxy for

our claim on "catastrophic forgetting":a longer output indicates that the SFT phase (verbose) has partially overwritten the RL phase (concise). The role of the history parameter $\alpha$ is to stabilize the identification of "RL-critical" parameters over time. For the 7B model, which has a stable training process because of its larger parameter capacity, the history mechanism ($\alpha > 0$) effectively accumulates long-term signals to identify truly critical RL parameters. This allows MIFO to freeze them robustly, preventing SFT overwriting and keeping the output concise (1067 tokens, closer to the RL baseline). MIFO$^{\dagger}$ ($\alpha = 0$) lacks this long-term stability, allowing more SFT verbosity to leak through (1371 tokens). Conversely, the 1.5B model is more volatile and unstable during training because of the limited parameter capacity. In this unstable regime, history can become noise: the accumulated importance map may lag behind the model's rapid shifts. Consequently, MIFO's historical mask is less precise for the small model, allowing SFT verbosity to influence the generation (2518 tokens). MIFO$^{\dagger}$, by relying solely on the most recent, immediate update, captures the volatile 1.5B model's current state more accurately, blocking SFT influence and retaining the short RL length (985 tokens). Note that despite the length difference, both methods achieve similar high accuracy, validating the general efficacy of the freezing approach.

