# OpenReview forum: "Mitigating Forgetting Between Supervised and Reinforcement Learning Yields Stronger Reasoners"
_ICLR.cc/2026/Conference — Submitted to ICLR 2026_

### Official Review · Reviewer_Qgmg · 2025-10-24

**Soundness:** 2
**Presentation:** 3
**Contribution:** 3
**Rating:** 2
**Confidence:** 3

**Summary:**

This paper introduces MIFO (Mitigating Forgetting between SFT and RL), a framework that dynamically integrates supervised fine tuning (SFT) with reinforcement learning (RL) to mitigate high data requirements and overfitting inherent to SFT while expanding reasoning frontiers through judicious use of out of distribution data. A highlight of MIFO is that it overcomes the phenomenon of catastrophic forgetting by-design. Experimental evaluations demonstrate that MIFO, and a variant MIFO$^+$ result in significantly improved data usage compared to state of the art.

**Caveat**: I am not fully familiar with the scope of mathematical reasoning benchmarks and datasets that are used in the evaluation (as mentioned by the authors in Sec. 5.1. My review is based on a presumption that these are adequate; I will defer to the authors/ other reviewers on whether any other benchmark/ dataset can potentially serve as additional datapoints for evaluation of MIFO.

**Strengths:**

(+) Interleaving of SFT with RL is a methodology that is unique relative to prior work. The merits of such an approach are revealed in the fact that it obviates a need for large amounts of data that is typically required for SFT.

(+) MIFO aims to mitigate catastrophic forgetting by design, which is significantly different from approaches in current art.

(+) The results demonstrating the redundancy in SFT relative to RL is particularly insightful, and serves as a strong basis for the working of MIFO in terms of mitigating catastrophic forgetting typical of SFT.

(+) The paper is generally well-written and the logical flow is sound. At the same times, I have some questions about the technical aspects and experiments (please see Weaknesses, below).

**Weaknesses:**

(-) The central claim by the authors is that MIFO is agnostic to the specific RL or SFT algorithm used (e.g., the claim that an RL algorithm different than GRPO can be used for RL training at the start of Sec. 4.1). However, the experimental evaluations do not seem to suggest that this claim has indeed been tested on using MIFO with multiple RL/ SFT algorithms.

(-) In Fig. 1 right, while the gap does begin to close at around the 110th step as the authors write, it subsequently begins to diverge. The text of the paper does not appear to provide an explanation for this phenomenon.

(-) In Fig. 1, while the gap between the SFT curves closes at the 40th step, and remains close subsequently and the gap between RL curves closes at the 110th step, it is not clear what other factors determine convergence. Also, from the right side of Fig. 1, it is not clear that RL converges even at 350 gradient steps.

(-) The labels on the graph of Fig. 2 do not seem to match with the caption of the figure or the text in Lines 159-161. Perhaps the green curve corresponds to SFT while the blue curve corresponds to RL?

(-) In Tables 1 and 2, it is not clear why MIFO produces shorter length outputs than MIFO$^+$ for the 7B model, while MIFO$^+$ yields significantly shorter length outputs than MIFO for the 1.5B model. Some insight into this result, and intuition about the role of the history parameter $\alpha$ will help make the interpretation of the results more clear.

(-) Some aspects of the presentation can be improved. For example, in Lines 269-270, the authors write `Entropy describes the uncertainty…’ - the writing will benefit from having a more formal definition of entropy over here.

(-) Minor comment: typo - in Line 194, there is an additional space between ( and Section.

**Questions:**

Please see Weaknesses above.

---

> ### Author Response · Authors · 2025-11-19
> **Response to Reviewer Qgmg [1/2]**
>
> Dear Reviewer Qgmg,
>
> Thank you for your constructive questions and helpful comments. Here we address your concerns as follows. We also integrate all updates into our revised manuscript.
>
> ---
>
> **Q1 MIFO is algorithm-agnostic but lacks the experiments with more RL/ SFT algorithms.**
>
>
> **A1** We appreciate the reviewer’s attention to this claim and agree that our experimental scope focused primarily on GRPO. We wish to clarify that our description of MIFO as "algorithm-agnostic" refers to its **structural compatibility** rather than an assertion of exhaustive empirical testing across all algorithms. MIFO treats the RL phase as a modular component that provides two generic outputs: rollout accuracy and entropy for data/token selection and parameter update magnitudes for the freezing mechanism. Since these two signals are inherent to *any* gradient-based reinforcement learning method (e.g., DAPO), the MIFO framework can mechanically integrate with them without modification to its core logic.
>
> Therefore, the validity of the proposed mechanism, which mitigates catastrophic forgetting by freezing parameters that changed significantly during the RL phase, holds regardless of the specific optimizer used to generate those changes. We utilized GRPO as the primary instantiation because it is one of the most popular methods for reasoning post-training. Besides, to address the reviewer's concern regarding the claim, we further provide additional experiments on multiple SFT and RL combinations [1,2] in Table P with sampled 15k training data for 1epoch training, which shows the consistent effectiveness of MIFO on improving different SFT-RL combinations.
>
> Table P. Performances with different SFT-RL algorithm combinations.
> |        | SFT+GRPO | DFT+GRPO | SFT+DAPO | DFT+DAPO |
> |--------|----------|----------|----------|----------|
> | SFT    | 23.6     | 23.1     | 23.6     | 23.1     |
> | RL     | 27.5     | 27.5     | 27.8     | 27.8     |
> | SFT-RL | 27.9     | 27.3     | 28.2     | 27.9     |
> | MIFO   | **30.7** | **29.2** | **30.5** | **29.7** |
>
>
> ---
> **Q2. In Fig. 1 right, the gap for RL curves closes at around the 110th step and then begins to diverge.**
>
> **A2** We thank the reviewer for pointing out the late-stage divergence in Figure 1 (Right). We agree with this observation and believe it further strengthens our central hypothesis that RL updates are parsimonious (non-redundant). While the "RL w. Gradient Drop" model initially catches up during the coarse learning phase (up to step 110), it subsequently hits a performance ceiling and diverges from the baseline. This indicates that as optimization requires finer adjustments, the random removal of 50\% of the gradient destroys essential information that RL. This divergence contrasts sharply with the SFT results, where the gradient-dropped model almost matches the baseline in the late training process, confirming SFT's high redundancy. The widening gap in the RL experiment effectively demonstrates an "learning bottleneck" caused by sparsification, proving the parsimonious updates of RL learning with less redundancy.

---

> ### Author Response · Authors · 2025-11-19
> **Response to Reviewer Qgmg [2/2]**
>
> **Q3. The differences in sparsity behavior in several steps in Figure 1.**
>
> **A3** We thank the reviewer for highlighting the differences in convergence behavior. The disparity in convergence speed and the factors stem from the difference between the two optimization objectives. SFT utilizes a strong, dense supervised signal (per-token cross-entropy), which naturally leads to rapid convergence (around step 40) and high redundancy. In contrast, RL optimization is inherently more complex and sample-inefficient due to the sparsity of the reward signal and the need for exploration on its own distribution; the model must discover optimal paths rather than just memorizing data from another distribution. This difficulty explains the RL curves close the gap slower (step 110) and why the "Gradient Drop" penalty is so severe: every update in RL is harder to acquire, and decision redundancy is less (Theorem 1), thus less redundant on parameter updates.
>
> Regarding the observation that RL has not fully converged at step 350, we have a response from two aspects: 1) We present  Post-hoc update sparsification results in Figure 14 for the model converged at step 1070 (Figure 12 confirms this convergence). Figure 14 still shows that SFT has more redundancy compared with RL at parameter updating at convergence. 2) We acknowledge that RL may require more training steps than SFT to reach complete saturation. However, we controlled for computational budget (epochs and steps) to ensure a fair comparison of *update redundancy*. Even if the RL baseline is still marginally improving at step 350, the key finding remains robust: the SFT model with 50% gradient drop is comparable to its baseline, whereas the RL model with gradient drop shows a distinct and persistent performance degradation. The fact that the RL baseline does not converge also underscores that its parameter updates contain non-redundant, necessary information that the sparsified model is missing.
>
> **Q4. length discrepancy of MIFO/MIFO+ for 1.5B and 7B model.**
>
> **A4** We thank the reviewer for this insightful observation. To interpret these results, it is crucial to first note the baseline behaviors: **SFT tends to be extremely verbose** (>10k tokens) while **RL encourages conciseness** (<1k tokens). Therefore, the output length serves as a proxy for "catastrophic forgetting": a longer output indicates that the SFT phase (verbose) has partially overwritten the RL phase (concise). The role of the history parameter $\alpha$ is to stabilize the identification of "RL-critical" parameters over time.
>
> For the **7B model** (Table 2), which has a stable training process because of its larger parameter capacity, the history mechanism ($\alpha > 0$) effectively accumulates long-term signals to identify truly critical RL parameters. This allows MIFO to freeze them robustly, preventing SFT overwriting and keeping the output concise (1067 tokens, closer to the RL baseline). $\text{MIFO}^\dagger$ ($\alpha=0$) lacks this long-term stability, allowing more SFT verbosity to leak through (1371 tokens).
> Conversely, the **1.5B model** (Table 1) is more volatile and unstable during training because of the limited parameter capacity. In this unstable regime, "history" can become "noise": the accumulated importance map may lag behind the model's rapid shifts. Consequently, MIFO's historical mask is less precise for the small model, allowing SFT verbosity to influence the generation (2518 tokens). $\text{MIFO}^\dagger$, by relying solely on the most recent, immediate update, captures the volatile 1.5B model's current state more accurately, blocking SFT influence and retaining the short RL length (985 tokens). Note that despite the length difference, both methods achieve similar high accuracy, validating the general efficacy of the freezing approach.
>
>
>
> **Q5. Typos and writing**
>
> **A5** Thank you for bringing this to our attention! We have fixed typos in Figure 2 and deleted the additional space accordingly in our manuscript. We also provided an entropy definition for tokens in line 273.
>
> ---
>
> **References**
>
> [1] On the Generalization of SFT: A Reinforcement Learning Perspective with Reward Rectification, arXiv'25
>
> [2] DAPO: An Open-Source LLM Reinforcement Learning System at Scale, arXiv'25

---

> > ### Comment · Reviewer_Qgmg · 2025-11-21
> > **Thank You Authors**
> >
> > I thank the authors for their detailed response to reviewer comments. I will increase my score by 2 points.

---

> > > ### Author Response · Authors · 2025-11-21
> > > **Thank You!**
> > >
> > > We thank you for reviewing our rebuttal and for raising the score. We are glad that our response resolved your concerns. The revised manuscript has been updated to include these discussions (highlighted in blue). Thank you again for your time and constructive feedback

---

> ### Author Response · Authors · 2025-11-21
> **Additional Response to Reviewer Qgmg**
>
> **Q6. MIFO's effectiveness on other domains or other non-math models**
>
> **A6.** We also apologize that we missed the additional experiments on other benchmarks in our first round response, as mentioned in your summary caveat. Regarding the generalizability of MIFO, we expand our evaluation beyond the initial knowledge reasoning results on MMLU-Pro (Tables 1 and 2). We now provide additional experimental results covering code generation, scientific/knowledge-based reasoning, and different model backbones (detailed setting in Appendix E.6). Specifically, Table Q demonstrates MIFO’s superiority on code generation tasks using Qwen2.5-7B-Instruct-1M. MIFO achieves the remarkable performance across all benchmarks. Notably, MIFO reaches 62.9% on average score, a 1.7% improvement over the SFT$\to$RL. Table R highlights consistent improvements in math reasoning using the Llama 3.2 8B model. MIFO outperforms the competitive SFT+RL baseline ReLIFT by 2.4 points on average. Finally, Table S confirms MIFO's effectiveness on scientific and knowledge-based benchmarks. MIFO improves over the SFT$\to$RL baseline by a massive 17 points on average. MFIO performs comparably to SRFT on the average score with much less data usage, presented in Section 5.2. Collectively, these extensive experiments validate MIFO as a versatile solution for the SFT-RL forgetting problem, demonstrating that it can robustly generalize to deliver consistent gains across varying model sizes, architectures, and complex reasoning domains.
>
> Table Q. Comparison of coding task performances.
> |                     | LCB      | HumanEval+ | MBPP+    | Avg      |
> |---------------------|----------|------------|----------|----------|
> | Qwen2.5-7B-Instruct | 24.0     | 80.5       | 66.7     | 57.1     |
> | SFT                 | 24.5     | 81.7       | 66.9     | 57.7     |
> | RL                  | 28.6     | 84.8       | 70.1     | 61.2     |
> | SFT-RL              | 28.8     | 83.5       | 71.4     | 61.2     |
> | MIFO                | **29.1** | **86.6**   | **73.0** | **62.9** |
>
> Table R. Comparison of reasoning results for Llama3.2 8B model.
> |        |  AIME24 |  AIME25 |    AMC   |   MATH   | Olympiad | MMLU-Pro |    Avg   |
> |--------|:-------:|:-------:|:--------:|:--------:|:--------:|:--------:|:--------:|
> | SFT    |   0.8   |   1.5   |   11.5   |   28.6   |    8.6   |   28.2   |   13.2   |
> | RL     |   1.8   |   0.0   |   10.8   |   28.2   |    7.3   |   39.4   |   14.6   |
> | ReLIFT |   1.3   |   0.2   |   11.9   |   35.2   |   11.0   |   44.2   |   17.3   |
> | MIFO   | **2.1** | **2.5** | **13.9** | **40.6** | **12.3** | **46.6** | **19.7** |
>
> Table S.  Comparison of scientific/knowledge-based reasoning results.
> |        | ARC-c    | GPQA     | MMLU-Pro | Avg      |
> |--------|----------|----------|----------|----------|
> | Base   | 70.3     | 24.7     | 34.1     | 43.0     |
> | SFT    | 75.2     | 24.7     | 42.7     | 47.5     |
> | RL     | 83.2     | 40.4     | 49.3     | 57.6     |
> | SFT-RL | 72.4     | 24.2     | 37.7     | 44.8     |
> | LUFFY  | 80.5     | 39.9     | 52.6     | 57.7     |
> | SRFT   | **85.3** | 46.4     | 54.1     | **61.9** |
> | MIFO   | 83.8     | **47.4** | **54.3** | 61.8     |

---

### Official Review · Reviewer_jiz5 · 2025-10-28

**Soundness:** 3
**Presentation:** 2
**Contribution:** 2
**Rating:** 4
**Confidence:** 3

**Summary:**

This paper proposes MIFO, a plug-and-play framework to jointly optimize SFT and RL for reasoning post-training of LLMs. The key claim is that SFT introduces redundant and high-magnitude parameter updates that overwrite the more updates of RL, leading to catastrophic forgetting. To address this, MIFO Interleaves SFT into RL, selecting only challenging rollouts and applying loss only on high-entropy tokens. MIFO achieves perfect results on AIME-24/25, AMC, MATH-500, OlympiadBench, and MMLU-Pro, while using only 1.5% of the SFT data and 20.4% of the RL data.

**Strengths:**

1. Identifies and visualizes the gradient update magnitude between SFT and RL.

2. Consistently gains across different reasoning benchmarks.

3. Solid ablations of complementary effects of entropy-based token selection and parameter freezing.

**Weaknesses:**

1. The freezing and entropy ideas, while effective, are incremental extensions of existing interleaved SFT+RL frameworks (e.g., ReLIFT).

2. All experiments use Qwen-Math models on mathematical reasoning; no evidence of generalization to other domains or other model settings.

3. Limited discussion on compute or runtime overheads.

**Questions:**

1. How sensitive is MIFO to the hyperparameters?

2. Have you tested MIFO with other domains or other non-math models?

3. What is the computational overhead (e.g., GPU hours) of MIFO compared to baselines?

4. Theoretical analysis (Appendix C) is disconnected from practice. The introduced Decision–Redundancy Ratio (DR) is not computed empirically nor related to the actual experiments in main sections. It’s unclear what the analysis truly verifies.

5. Forgetting not quantitatively measured. Figures only visualize parameter update magnitude, not actual forgetting metrics. The claim that MIFO “mitigates forgetting” is weakly supported.

6. The paper lacks any figure showing model performance over training steps (e.g., test dataset performances across training steps). Such curves would clarify whether MIFO actually stabilizes learning rather than just improving final accuracy.

---

> ### Author Response · Authors · 2025-11-19
> **Response to Reviewer jiz5 [1/3]**
>
> Dear Reviewer jiz5,
>
> We sincerely appreciate your professional and insightful suggestions for our work. We have taken your suggestions into account and addressed them below:
>
> ---
>
> **Q1 The freezing and entropy ideas, while effective, are incremental extensions of existing interleaved SFT+RL frameworks (e.g., ReLIFT).**
>
> **A1** We want to clarify again that MIFO's central idea of mitigating forgetting is completely different with previous methods. The only same idea is interleaving SFT+RL, and in section 4.1 and Appendix F.3, we have explained the detailed differences between ReLIFT and MIFO.  Actually, our main contribution is the first to identify the forgetting between SFT and RL training, and providing the corresponding solution with two main contribution to mitigate the forgetting 1) data processing for SFT+RL interleaved training 2) parameter tracking and freezing. the extensions for interleaved SFT+RL is only one part of our first contribution. Besides, the interleaving is also different as MIFO futher provides a scalable threshold, indicates that our subsequent modules explicitly mitigate forgetting to enlarge the usable data pool, and maximize the learning on existing knowledge of data to improve reasoning capability. Table I provides a summarized compairison.
>
> Table I. The summarized comparison with existing representive interleaved SFT+RL frameworks ReLIFT.
> |        | Interleave SFT+RL | Threshold for Informative Sample Selection | Token Selection for Tokens | Parameter Freezing | Mitigating Forgetting | Large Performance Gain Compared with SFT-> RL |
> |--------|-------------------|--------------------------------------------|----------------------------|--------------------|-----------------------|-----------------------------------------------|
> | ReLIFT | Yes               | No                                         | No                         | No                 | No                    | No (+0.5 for 3B, +1.3 for 7B)                   |
> | MIFO   | Yes               | Yes                                        | Yes                        | Yes                | Yes                   | Yes (+6.5 for 3B, +5.0 for 7B)                  |

---

> ### Author Response · Authors · 2025-11-19
> **Response to Reviewer jiz5 [2/3]**
>
> **Q2. MIFO on other domains or other non-math models**
>
> **A2.** Regarding the generalizability of MIFO, we expand our evaluation beyond the initial knowledge reasoning results on MMLU-Pro (Tables 1 and 2). We now provide additional experimental results covering code generation, scientific/knowledge-based reasoning, and different model backbones (detailed setting in Appendix E.6). Specifically, Table K demonstrates MIFO’s superiority on code generation tasks using Qwen2.5-7B-Instruct-1M. MIFO achieves the remarkable performance across all benchmarks. Notably, MIFO reaches 62.9% on average score, a 1.7% improvement over the SFT$\to$RL. Table L highlights consistent improvements in math reasoning using the Llama 3.2 8B model. MIFO outperforms the competitive SFT+RL baseline ReLIFT by 2.4 points on average. Finally, Table M confirms MIFO's effectiveness on scientific and knowledge-based benchmarks. MIFO improves over the SFT$\to$RL baseline by a massive 17 points on average. And MFIO performs comparably to SRFT on the average score with much less data usage, presented in Section 5.2. Collectively, these extensive experiments validate MIFO as a versatile solution for the SFT-RL forgetting problem, demonstrating that it can robustly generalize to deliver consistent gains across varying model sizes, architectures, and complex reasoning domains.
>
> Table K. Comparison of coding task performances.
> |                     | LCB      | HumanEval+ | MBPP+    | Avg      |
> |---------------------|----------|------------|----------|----------|
> | Qwen2.5-7B-Instruct | 24.0     | 80.5       | 66.7     | 57.1     |
> | SFT                 | 24.5     | 81.7       | 66.9     | 57.7     |
> | RL                  | 28.6     | 84.8       | 70.1     | 61.2     |
> | SFT-RL              | 28.8     | 83.5       | 71.4     | 61.2     |
> | MIFO                | **29.1** | **86.6**   | **73.0** | **62.9** |
>
> Table L. Comparison of reasoning results for Llama3.2 8B model.
> |        |  AIME24 |  AIME25 |    AMC   |   MATH   | Olympiad | MMLU-Pro |    Avg   |
> |--------|:-------:|:-------:|:--------:|:--------:|:--------:|:--------:|:--------:|
> | SFT    |   0.8   |   1.5   |   11.5   |   28.6   |    8.6   |   28.2   |   13.2   |
> | RL     |   1.8   |   0.0   |   10.8   |   28.2   |    7.3   |   39.4   |   14.6   |
> | ReLIFT |   1.3   |   0.2   |   11.9   |   35.2   |   11.0   |   44.2   |   17.3   |
> | MIFO   | **2.1** | **2.5** | **13.9** | **40.6** | **12.3** | **46.6** | **19.7** |
>
> Table M.  Comparison of scientific/knowledge-based reasoning results.
> |        | ARC-c    | GPQA     | MMLU-Pro | Avg      |
> |--------|----------|----------|----------|----------|
> | Base   | 70.3     | 24.7     | 34.1     | 43.0     |
> | SFT    | 75.2     | 24.7     | 42.7     | 47.5     |
> | RL     | 83.2     | 40.4     | 49.3     | 57.6     |
> | SFT-RL | 72.4     | 24.2     | 37.7     | 44.8     |
> | LUFFY  | 80.5     | 39.9     | 52.6     | 57.7     |
> | SRFT   | **85.3** | 46.4     | 54.1     | **61.9** |
> | MIFO   | 83.8     | **47.4** | **54.3** | 61.8     |
>
> ---
> **Q3 Discussion on computational overheads.**
>
> **A3.** Theoretically, the computational complexity remains constant between stages, as the model architecture and forward/backward pass mechanisms are identical for both SFT and RL. Therefore, the only external overhead is tracking the parameter change. We present the GPU hours in Table N, which shows MIFO consumes a comparable training time compared with previous SOTA methods.
>
> Table N. The GPU hours for different methods.
> |          | SFT | RL   | SFT->RL | LUFFY | LUFFY+ | ReLIFT | MIFO |
> |----------|-----|------|---------|-------|--------|--------|------|
> | GPU Hour | 8x8 | 40x8 | 67x8    | 77x8  | 130x8  | 52x8   | 74x8 |

---

> ### Author Response · Authors · 2025-11-19
> **Response to Reviewer jiz5 [3/3]**
>
> **Q4. It’s unclear what introduced Decision–Redundancy theory truly verifies.**
>
> **A4.** We respectfully clarify that the theoretical analysis is not disconnected; rather, it provides the causal mechanism for the empirical observations in Section 3.1. The Decision-Redundancy Ratio (DR) theoretically quantifies how much a parameter update exceeds the minimal necessary change to adjust the decision margin of logits for the right token prediction. This concept directly explains *why* SFT exhibits "sparsification robustness": the high redundancy defined in Lemma 1 implies the redundancy to predict the right token after the gradient descent, allowing SFT to maintain performance even when 50% of updates are pruned (Figure 1, 2), whereas RL’s low redundancy (parsimonious updates) leads to immediate degradation under similar pruning. Since computing the exact DR is computationally intractable for LLMs, the sparsification experiments serve as the empirical proxy to validate Theorem 1. The theorem analytically predicts that $\mathrm{DR}^{\mathrm{SFT}} \ge \mathrm{DR}^{\mathrm{RL}}$, and this inequality is strictly confirmed by our experimental results showing SFT's better tolerance to pruning compared to RL. Following your suggestion, we have revised the text to explicitly state that the sparsification experiments act as the empirical validation of the theoretical DR predictions as follows.
>
> >Our theoretical analysis, formalized in Lemma 1 and Theorem 1, provides a causal mechanism for the empirical phenomena observed in Section 3.1. The core theoretical finding is that the Decision-Redundancy (DR) on logits for predicting the target token of the SFT is intrinsically higher than that of the RL after parameter updates (i.e., $\mathrm{DR}^{\mathrm{SFT}} \ge \mathrm{DR}^{\mathrm{RL}}$). This conclusion directly explains why SFT exhibits \textit{sparsification robustness}: Theorem 1 predicts SFT has a high DR, implying its parameter updates are "redundant." This redundancy allows SFT to maintain performance even when 50% of updates are pruned (as shown in Figures 1 and 2), as it has sufficient "wasteful" updates to secure the correct token prediction. Conversely, RL's low DR suggests its updates are \textit{parsimonious} and efficient, which explains its fragility and immediate performance degradation under similar pruning conditions. Since calculating the exact DR is computationally intractable for large language models, our sparsification experiments serve as the empirical proxy to validate this theory. Theorem 1 predicts this imbalance, and the experiments (Figures 1 and 2) strictly confirm it by demonstrating SFT's better tolerance to pruning.
> >
> ---
>
> **Q5. Forgetting is not quantitatively measured.**
>
> **A5.** To quantitatively measure the forgetting, we design an experiment to fairly measure the forgetting condition for different baselines and MIFO. We first train the model with RL for 50 steps (average test score=24.3 at the end), and then do training for multiple methods containing SFT objectives for 10 steps for evaluation. Table O shows the final test accuracy after training: if the training objective has SFT, the model performances largely drop after the previous RL training, indicating the catastrophic forgetting caused by SFT, while MIFO experiences a performance gain after training, indicating its effectiveness on mitigating forgetting.
>
> Table O. The quantitatively measured forgetting.
> |        | RL   | SFT   | LUFFY  | SRFT | ReLIFT | MIFO |
> |--------|------|-------|-------|-------|--------|------|
> | Final  | 25.4 | 8.7   | 11.7  | 15.7  | 13.2   | 25.2 |
> | Change | +1.1  | -15.6 | -12.6 | -8.6  | -11.1   | +0.9 |
>
> Also, our ablation study in Section 5.3 provides another straight evidence on "mitigates forgetting". Comparing +Interleave and +Interleave+PF, we find that just freezing a large amount of parameters for SFT training not only avoids performance drop, but also has external markable improvements (+4.4). If our claim on mitigating forgetting is wrong, the performance wouldn't gain a large improvement after parameter freezing.
>
> ---
>
> **Q6. Lacks figure showing model performance over training steps.**
>
> **A6** We have provided the model performances of MIFO and ablated models, including average validation score on test datasets, batch solved rate, batch all solved rate, average reward value, and over training steps in Appendix E.3. These curves showing training dydanmics clearly illustrated the MIFO provides a stablized learning from accuracy, reward and batch solving perspectives, and the effectness of all proposed modules.

---

### Official Review · Reviewer_p9ox · 2025-10-30

**Soundness:** 2
**Presentation:** 3
**Contribution:** 2
**Rating:** 4
**Confidence:** 4

**Summary:**

This paper proposes MIFO, an interleaved SFT RL post training framework to mitigate forgetting. MIFO mainly consists of two components: data processing to strengthen low accuracy examples for SFT, and parameter freezing to prevent overwriting key parameters.

**Strengths:**

- MIFO outperforms multiple baselines on math tasks.
- MIFO improves data efficiency than baselines.

**Weaknesses:**

Weaknesses:
- The experimental validations mainly focus on math tasks, while other reasoning tasks beyond math are overlooked.
- Experiments focus on Qwen family, making applicability of MIFO to other model families unclear especially given the observed performance drop under different templates.
- MIFO relies on experts or a stronger teacher model. The cost associated with it is ignored in validations.
- Linearized approximation in theoretical analysis in Appendix C needs to be justified.
- Results in Figure 2 seem contradicting with description in lines 160-161. Please clarify.

**Questions:**

See Weaknesses

---

> ### Author Response · Authors · 2025-11-19
> **Response to Reviewer p9ox [1/2]**
>
> Dear Reviewer p9ox,
>
> We sincerely appreciate your professional and insightful suggestions for our work. We have taken your suggestions into account and addressed them below:
>
> ---
>
> **Q1** Other reasoning tasks beyond math, and other models.
>
> **A1** Regarding the generalizability of MIFO, we expand our evaluation beyond the initial knowledge reasoning results on MMLU-Pro (Tables 1 and 2). We now provide additional experimental results covering code generation, scientific/knowledge-based reasoning, and different model backbones (detailed setting in Appendix E.6). Specifically, Table F demonstrates MIFO’s superiority on code generation tasks using Qwen2.5-7B-Instruct-1M. MIFO achieves the remarkable performance across all benchmarks. Notably, MIFO reaches 62.9% on average score, a 1.7% improvement over the SFT$\to$RL. Table G highlights consistent improvements in math reasoning using the Llama 3.2 8B model. MIFO outperforms the competitive SFT+RL baseline ReLIFT by 2.4 points on average. Finally, Table H confirms MIFO's effectiveness on scientific and knowledge-based benchmarks. MIFO improves over the SFT$\to$RL baseline by a massive 17 points on average. MFIO performs comparably to SRFT on the average score with much less data usage, presented in Section 5.2. Collectively, these extensive experiments validate MIFO as a versatile solution for the SFT-RL forgetting problem, demonstrating that it can robustly generalize to deliver consistent gains across varying model sizes, architectures, and complex reasoning domains.
>
> Table F. Comparison of coding task performances.
> |                     | LCB      | HumanEval+ | MBPP+    | Avg      |
> |---------------------|----------|------------|----------|----------|
> | Qwen2.5-7B-Instruct | 24.0     | 80.5       | 66.7     | 57.1     |
> | SFT                 | 24.5     | 81.7       | 66.9     | 57.7     |
> | RL                  | 28.6     | 84.8       | 70.1     | 61.2     |
> | SFT-RL              | 28.8     | 83.5       | 71.4     | 61.2     |
> | MIFO                | **29.1** | **86.6**   | **73.0** | **62.9** |
>
> Table G. Comparison of reasoning results for Llama3.2 8B model.
> |        |  AIME24 |  AIME25 |    AMC   |   MATH   | Olympiad | MMLU-Pro |    Avg   |
> |--------|:-------:|:-------:|:--------:|:--------:|:--------:|:--------:|:--------:|
> | SFT    |   0.8   |   1.5   |   11.5   |   28.6   |    8.6   |   28.2   |   13.2   |
> | RL     |   1.8   |   0.0   |   10.8   |   28.2   |    7.3   |   39.4   |   14.6   |
> | ReLIFT |   1.3   |   0.2   |   11.9   |   35.2   |   11.0   |   44.2   |   17.3   |
> | MIFO   | **2.1** | **2.5** | **13.9** | **40.6** | **12.3** | **46.6** | **19.7** |
>
> Table H.  Comparison of scientific/knowledge-based reasoning results.
> |        | ARC-c    | GPQA     | MMLU-Pro | Avg      |
> |--------|----------|----------|----------|----------|
> | Base   | 70.3     | 24.7     | 34.1     | 43.0     |
> | SFT    | 75.2     | 24.7     | 42.7     | 47.5     |
> | RL     | 83.2     | 40.4     | 49.3     | 57.6     |
> | SFT-RL | 72.4     | 24.2     | 37.7     | 44.8     |
> | LUFFY  | 80.5     | 39.9     | 52.6     | 57.7     |
> | SRFT   | **85.3** | 46.4     | 54.1     | **61.9** |
> | MIFO   | 83.8     | **47.4** | **54.3** | 61.8     |
>
> ---
>
> **Q2. MIFO relies on experts or a stronger teacher model. The cost is ignored in validations.**
>
> **A2** Current standard SFT relies on high-quality SFT data from experts' annotation or stronger teacher models. This is the intrinsic requirement of SFT, and any method can't avoid this cost as long as it incorporates SFT. Besides, our MIFO largely saves the SFT data cost to 1.5% the previous SOTA, and we provide the detailed comparison on SFT data cost in Section 5.2.

---

> ### Author Response · Authors · 2025-11-19
> **Response to Reviewer p9ox [2/2]**
>
> **Q3. Linearized approximation in theoretical analysis needs to be justified.**
>
> **A3** We agree that clarifying this assumption is important. Our analysis uses a first-order Taylor approximation to analyze the *local* change in the decision margin, $m(\theta)$, with respect to a small parameter update, $\Delta$. This local linearization is a standard and fundamental assumption in the analysis of deep learning optimization [1,2]. In fact, it is the same principle that justifies gradient descent itself. When we compute a gradient $\nabla L(\theta)$, we are implicitly assuming that the loss function $L$ is locally linear. This local approximation is precisely what guarantees a decrease in loss for a sufficiently small learning rate $\eta$ [2].
> Our goal with this lemma is to quantify the *local efficiency* of a single parameter update ($\Delta\theta$) by comparing it to the ideal "shortest path" update ($\Delta^\star$) derived from this local geometry. Given that the logit margin function $m(\theta)$ is a smooth, differentiable function with respect to the parameters $\theta$, this Taylor approximation is a valid and standard tool for analyzing its behavior in a small neighborhood around the pre-update parameters $\theta_0$.
> We will add a sentence to Appendix C to explicitly state that this is a first-order Taylor approximation, which is standard for analyzing the local geometry of optimization landscapes.
>
> ---
>
> **Q4** Typos in Figure 2.
>
> **A4** Thank you for bringing this to our attention! We have fixed typos in Figure 2 accordingly in our manuscript.
>
> ---
> **References**
>
> [1] Convex Optimization. Cambridge University Press, 2004
>
> [2] Optimization methods for large-scale machine learning. SIAM Review, 2018

---

> > ### Comment · Reviewer_p9ox · 2025-11-28
> >
> > I appreciate the authors' time and effort in addressing the comments. The additional experiments strengthen the contribution. I will raise my score.

---

### Official Review · Reviewer_RRa6 · 2025-10-30

**Soundness:** 3
**Presentation:** 3
**Contribution:** 3
**Rating:** 6
**Confidence:** 3

**Summary:**

The work proposes MIFO, Mitigating Forgetting Between SFT and RL, a new pipeline to bridge the SFT and RL in post-training of LLM reasoning. The pipeline starts from RL and constructs an SFT data buffer. Then, it uses entropy-based token selection and RL parameter update-based freezing for SFT. The comprehensive experiments on small-scale LLMs show that it can efficiently boost the model's reasoning performance.

**Strengths:**

- It proposed an interesting view of SFT and RL in post-training of LLM reasoning. The analysis in section 3 provided good motivation for the design of MIFO. And the components of MIFO provide a promising advantage to improve the training of SFT+RL.
- The experiment as well as ablation study, indicates MIFO is well effective compared to baseline, and provides good data efficiency and token efficiency.

**Weaknesses:**

- The experiment is purely based on qwen 2.5 models and the math domain training dataset. The generalizability of this approach to other domain is tricky. And qwen 2.5 models (even it is the base model) include heavy mid-training data, experiment on these models are more like containing an implicit SFT, which is different from the claimed RL-first-then-SFT paradiam.
- Risk of catastrophic forgetting is mentioned as motivation, but not studied/showed how MIFO addressed this.
- Writing issue: e.g., L323 NuminaMath Li et al. (2024) > NuminaMath (Li et al., 2024)

**Questions:**

- Though I did not see the code, I think the proposed method will introduce computation overhead due to the (frequent) context switch between SFT and RL. As my question below, the interval matters in this design to balance training performance and training efficiency.
- I did not find out MIFO iteration number/interval used for the experiment. And what is the effect of these factor? For example, it can be high high-frequency interval (e.g., every batch), or a low-frequency interval (e.g, every epoch).
- I am interested in the data buffer dynamics of training. In other words, does the effective buffer get smaller during training, and does the questions get back and forth in the buffer? This information helps the understanding of whether the model learns something new to improve the performance, otherwise it may be more like randomness.  Also, I wonder how often the frozen RL updated parameters overlap with those from high-entropy sft tokens. My intuition is that these are pretty much overlapped, so I did not understand what is updated.

---

> ### Author Response · Authors · 2025-11-19
> **Responses to Reviewer RRa6 [1/3]**
>
> Dear Reviewer RRa6,
>
> Thank you for recognizing the contributions and analysis offered by our work, as well as for the constructive suggestions! We have addressed all your comments and suggestions as follows.
>
> ----
>
> **Q1. Qwen 2.5 models  include heavy mid-training data containing an implicit SFT, which is different from the claimed RL-first-then-SFT paradiam.**
>
>
>
> **A1** We respectfully clarify that our interleaved "RL-SFT" paradigm refers specifically to the **post-training optimization order** applied to the starting checkpoint, not the pre-training history of that checkpoint. While we acknowledge that modern "Base" models like Qwen 2.5 undergo extensive mid-training, this actually strengthens our findings: despite the model's robust initial capabilities, we still observe the specific phenomenon of catastrophic forgetting in the SFT+RL post-training stage, where subsequent standard SFT overwrites the gains made by new RL exploration. The "implicit SFT" in the base model is a constant across all our baselines, and our contribution is demonstrating that *given* a strong starting point, previous practices of applying combined SFT+RL is suboptimal compared to MIFO, which protects the delicate, newly-acquired reasoning paths found during the RL phase from SFT, saving large amount of training data at the same time.
>
> ---
> **Q2. Experiments on more domains and models.**
>
> **A2.** Regarding generalizability of MIFO, we expand our evaluation beyond the initial knowledge reasoning results on MMLU-Pro (Tables 1 and 2). We now provide additional experimental results covering code generation, scientific/knowledge-based reasoning, and different model backbones (detailed setting in Appendix E.6). Specifically, Table A demonstrates MIFO’s superiority on code generation tasks using Qwen2.5-7B-Instruct-1M. MIFO achieves the remarkable performance across all benchmarks. Notably, MIFO reaches 62.9% on average score, a 1.7% improvement over the SFT$\to$RL. Table B highlights consistent improvements in math reasoning using the Llama 3.2 8B model. MIFO outperforms the compatitive SFT+RL baseline ReLIFT by 2.4 points on average. Finally, Table C confirms MIFO's effectiveness on scientific and knowledge-based benchmarks. MIFO improves over the SFT$\to$RL baseline by a massive 17 points on average. And MFIO performs comparably to SRFT on the average score with much less data usage, presented in Section 5.2. Collectively, these extensive experiments validate MIFO as a versatile solution for the SFT-RL forgetting problem, demonstrating that it can robustly generalize to deliver consistent gains across varying model sizes, architectures, and complex reasoning domains.
>
> Table A. Comparison on coding task performances.
> |                     | LCB      | HumanEval+ | MBPP+    | Avg      |
> |---------------------|----------|------------|----------|----------|
> | Qwen2.5-7B-Instruct | 24.0     | 80.5       | 66.7     | 57.1     |
> | SFT                 | 24.5     | 81.7       | 66.9     | 57.7     |
> | RL                  | 28.6     | 84.8       | 70.1     | 61.2     |
> | SFT$\to$RL              | 28.8     | 83.5       | 71.4     | 61.2     |
> | MIFO                | **29.1** | **86.6**   | **73.0** | **62.9** |
>
> Table B. Comparison on math reasoning results for Llama3.2 8B.
> |        |  AIME24 |  AIME25 |    AMC   |   MATH   | Olympiad | MMLU-Pro |    Avg   |
> |--------|:-------:|:-------:|:--------:|:--------:|:--------:|:--------:|:--------:|
> | SFT    |   0.8   |   1.5   |   11.5   |   28.6   |    8.6   |   28.2   |   13.2   |
> | RL     |   1.8   |   0.0   |   10.8   |   28.2   |    7.3   |   39.4   |   14.6   |
> | ReLIFT |   1.3   |   0.2   |   11.9   |   35.2   |   11.0   |   44.2   |   17.3   |
> | MIFO   | **2.1** | **2.5** | **13.9** | **40.6** | **12.3** | **46.6** | **19.7** |
>
> Table C.  Comparison on scientific/knowledge based reasoning results.
> |        | ARC-c    | GPQA     | MMLU-Pro | Avg      |
> |--------|----------|----------|----------|----------|
> | Base   | 70.3     | 24.7     | 34.1     | 43.0     |
> | SFT    | 75.2     | 24.7     | 42.7     | 47.5     |
> | RL     | 83.2     | 40.4     | 49.3     | 57.6     |
> | SFT$\to$RL | 72.4     | 24.2     | 37.7     | 44.8     |
> | LUFFY  | 80.5     | 39.9     | 52.6     | 57.7     |
> | SRFT   | **85.3** | 46.4     | 54.1     | **61.9** |
> | MIFO   | 83.8     | **47.4** | **54.3** | 61.8     |

---

> ### Author Response · Authors · 2025-11-19
> **Responses to Reviewer RRa6 [2/3]**
>
> **Q3. Lack of show on how MIFO addressed the catastrophic forgetting.**
>
> **A3.** We respectfully clarify that mitigating catastrophic forgetting is a core design principle of MIFO, explicitly studied and addressed through two distinct mechanisms in the paper. We first investigated the root cause in **Section 3.2 (SFT Forgets RL)**, providing empirical evidence in **Figure 3** that SFT induces significantly larger parameter update magnitudes than RL, which leads to the overwriting of parsimonious RL updates. To address this at the data level, **Section 4.1** details our sample and high-entropy token selection strategy, which restricts SFT parameter updates with smaller magnitudes, preventing forgetting on RL updates. Furthermore, we propose a direct algorithmic intervention to mitigate forgetting, formulated in **Section 4.2 (Parameter Freezing)**. MIFO freezes RL-critical parameter updates and explicitly limits the SFT parameter update. We also summarize how MIFO addresses forgetting inthe  introduction as follows:
>
> > Most importantly, MIFO is designed with a core principle to solve \textit{(iii)} catastrophic forgetting. Our evidence shows that forgetting arises when extensive SFT updates overwrite the information acquired through RL. We therefore design MIFO with two complementary mechanisms. \textit{First}, the aforementioned data and token selection strategy not only enables MIFO plug-and-play with reduced data usage, but also limits the magnitude of SFT updates, which decreases the forgetting of the knowledge learned by previous RL. \textit{Second}, our analysis reveals an asymmetry between SFT and RL parameter updates. SFT tends to update parameters redundantly, so removing part of its updates does not harm performance significantly. RL, on the other hand, updates parameters in a more parsimonious way, and omitting part of its updates leads to clear performance degradation. Based on this observation, MIFO dynamically identifies RL-critical parameters, freezes them during SFT, and unfreezes them in the subsequent RL step. This design protects important RL parameter updates from being overwritten by SFT, effectively mitigating catastrophic forgetting.
>
> To quantify how MIFO mitigates forgetting, Appendix E.5 also presents an experiment to fairly measure the forgetting condition, confirming that MIFO can mitigate the forgetting caused by SFT well while other baselines can't.
>
> **Q4 The computation overhead due to the context switch between SFT and RL.**
>
> **A4** We appreciate this practical observation regarding training efficiency. Theoretically, the computational complexity (FLOPs) remains constant between stages, as the model architecture and forward/backward pass mechanisms are identical for both SFT and RL when they are integrated in MIFO. The "context switch" in our implementation is designed as a seamless logical transition within the training loop: MIFO simply swaps the data iterator and loss function rather than a physical reloading of model weights. Therefore, the only overhead is tracking the parameter instead of the context shift. We present the GPU hours in Table D, which shows MIFO consumes a comparable training time compared with previous SOTA methods.
>
> Table D. The GPU hours for different methods.
> |          | SFT | RL   | SFT->RL | LUFFY | LUFFY+ | ReLIFT | MIFO |
> |----------|-----|------|---------|-------|--------|--------|------|
> | GPU Hour | 8x8 | 40x8 | 67x8    | 77x8  | 130x8  | 52x8   | 74x8 |
>
> ---
>
> **Q5. What is the effect of MIFO iteration number? How's this dynamic**
>
> **A5** The iteration number is determined by the data size of the buffer (mentioned in Section 4.1), and this size is set to 64. If the buffer is full, RL shifts to SFT, and then SFT shifts to RL once training is finished using the data in the buffer. This buffer size is a fixed number during the training. This design aligns with our idea on "mitigating forgetting": a larger buffer size caused more forgetting that MIFO can't handle well, while a small buffer size can't have a comprehensive track of SFT updates and caused more overhead on shift. We provide a dynamic study on the effect of buffer size in Table E.
>
> MIFO buffer doesn't get smaller during training, and questions don't get back and forth in the buffer. When SFT using buffer data is done, the buffer will be cleared; therefore, the buffer will always hold new critical data waiting to be trained for SFT, improving performance.
>
>
> Table E. The average pass@1 test accuracy for different buffer sizes for Qwen2.5-1.5B-Math-Instruct, trained on 1 epoch.
> | Buffer Size   |   8  |  16  |  64  |  128 |  256 |
> |-----|:----:|:----:|:----:|:----:|:----:|
> | Avg | 32.4 | 34.4 | **36.3** | 36.1 | 35.8 |

---

> ### Author Response · Authors · 2025-11-19
> **Responses to Reviewer RRa6 [3/3]**
>
> **Q6.  How often the frozen RL updated parameters overlap with those from high-entropy sft tokens and the intuition is that these are pretty much overlapped. What is updated?**
>
> **A6.** We appreciate this insightful question, as it allows us to clarify a critical distinction. Your intuition that SFT parameter updates w. high-entropy tokens and RL-critical parameter updates might be related is plausible, but our method is designed to *decouple* them. The core clarification is that **high-entropy SFT tokens are the *source* of the loss signal** (where gradients originate), while **frozen RL-important parameters are included in the *target* of the SFT update** (where gradients are blocked, however). For a single RL-SFT interval, the loss calculated at SFT high-entropy tokens backpropagates gradients to *all* parameters in its computational path, not just a specific subset. You are correct that the overlap should be *caused* by the SFT, but our method's goal is to decrease this overlap on RL-critical parameters by only updating *other* parameters.
>
> This mechanism is the central goal of MIFO. When we freeze the RL-critical parameters (using mask $\mathbf{M}_i$), the gradients from high-entropy tokens are zeroed out for that subset, but they **still flow freely to and update the entire complementary set of this subset**. This relies on our finding in Section 3.1 that SFT has a "redundant" parameter space that can absorb new knowledge. By freezing the "parsimonious" RL parameter set, we *force* the SFT update (carries the new external knowledge) into this redundant space. In short, MIFO forces the model to learn new information from SFT by updating its non-critical components, successfully preserving the reasoning capability built by RL.
>
> ---
>
> **Q7** Typos
>
> **A7** Thank you for bringing this to our attention! We have fixed typos accordingly in our manuscript.

---

> > ### Comment · Reviewer_RRa6 · 2025-11-20
> >
> > Thanks for the clarification. I have a few follow-up questions.
> >
> > For Q1/Q3: While I think the clarification of RL-SFT paradigm is valid, I think the motivation for addressing 'catastrophic forgetting' is still vague. In RL stage, I think the model is hard to gain some new knowledge, but more on some meta-skill/pattern (aka reasoning ability). But a more straightforward 'catastrophic forgetting' issue is more related to domain shift, for example, you run RL(or SFT) on math domain, and then SFT on the coding domain, and come back to see the second SFT stage forget what it learned in the prior stage. That's what I mean, it lacks a study of how it addresses the issue. Also, in Deepseek-R1 report, they actually have the pipeline of two-cycle SFT-RL on base model, cold-start + RL for reasoning, then SFT + RL for the general domain. Such a paradigm is more related to the potential  'catastrophic forgetting' issue, and their success indicates the premise that it cannot run SFT after RL is not quite convincing.
> >
> > For Q4, as MIFO is not the best on efficiency, it's better to plot both performance and efficiency and show the Pareto Frontier.
> >
> > For Q6, it is related to my elaboration above, that when SFT needs to update critical knowledge, which may be on the frozon parameters determined by RL, the shift is bounded if they are the same domain, but it also makes the learning of new domain quite ineffective. And if it simply stays on the same domain, I think a straightforward baseline is to use smaller learning rate / using LoRA for SFT, but larger learning rate / full parameter update for RL.

---

> ### Author Response · Authors · 2025-11-21
> **Responses to Reviewer RRa6 [1/2]**
>
> Dear Reviewer RRa6,
>
> Thank you for acknowledging our response, and we further address your other concerns as follows:
>
> ---
>
> **Q8. The motivation of catastrophic forgetting.**
>
> **A8.** We thank you for being interested in detailed motivation and raising the distinction between classical catastrophic forgetting and the forgetting we study in this work. We will explain them in the following 4 aspects.
>
> **1) Scope and definition of forgetting**. We agree that traditional catastrophic forgetting is most commonly studied under explicit domain shift. Our work does not aim to resolve this well-studied setting, which is outside our scope. Instead, our goal is to identify and mitigate a hidden form of forgetting that arises between SFT and RL stages even when task domain is the same: a subsequent SFT stage can partially overwrite previous RL-induced behaviors for a same task distribution. Our study is the first to point out this risk, provide analysis and give a solution.
>
> **2) Regarding to DeepSeek-R1 and combined SFT+RL methods**. We appreciate the pointer to DeepSeek-R1’s two-cycle SFT+RL pipeline. **We do not claim that “you cannot run SFT after RL,” nor that such pipelines cannot succeed**. In fact, DeepSeek-R1, LUFFY, ReLIFT, SRFT and others clearly do succeed in terms of final performance. Our claim is more modest and complementary: these works demonstrate that combined SFT+RL can yield strong models, **but they do not measure whether and to what extent RL-induced improvements are partially forgotten in subsequent SFT phases.** Strong final metrics do not by themselves rule out the presence of degradation hidden in the reasoning training process caused by the forgetting phenomenon identified by our study. According to this motivation, MIFO further improves the combined SFT+RL performance by mitigating this degradation caused by forgetting.
>
> **3) Empirical evidence for motivation.** In Appendix E.5, we design targeted experiments to detect and quantify this phenomenon. Across all post-training methods that combine SFT and RL objectives, we observe a measurable drop in previous RL-improved performance after the methods' subsequent optimization steps (which contain SFT). This indicates that, even for pipelines that already achieve high-level performance (including DeepSeek-R1 and combined SFT+RL baselines), there remains headroom to preserve RL gains more faithfully. The parameter analysis in Section 3.2 provides an empirical explanation, and Appendix C provides a theoretical explanation for this forgetting.
>
> **4) Why our motivation is important**. Our proposed MIFO framework is precisely aimed at mitigating this hidden RL→SFT forgetting. The ablation in Section 5.3 also provides direct evidence: comparing +Interleave versus +Interleave+PF, we find that freezing a large subset of parameters during SFT not only avoids performance drops, but actually yields substantial additional gains (e.g., +4.4). If the information learned by RL were not being forgotten by SFT, such parameter freezing would be unlikely to produce a large and consistent improvement. Instead, the gains are exactly what we would expect if MIFO successfully protects RL-updated important parameters from being overwritten by subsequent SFT.
>
> ---
>
> **Q9. MIFO is not the best on training efficiency.**
>
> **A9.** Our claim is that MIFO achieves SOTA reasoning performance with strong data and response efficiency, but it is not specifically designed to optimize training efficiency. We appreciate this suggestion and will include a plot illustrating both performance and training efficiency, highlighting the Pareto frontier to better contextualize MIFO among competing methods.

---

> ### Author Response · Authors · 2025-11-21
> **Responses to Reviewer RRa6 [2/2]**
>
> **Q10.  It stays on the same domain with bounded shift, I think a straightforward baseline is to use smaller learning rate / using LoRA for SFT, but a larger learning rate / full parameter update for RL.**
>
> **A10.** Thank you for your insightful comment. We also provide a discussion on why utilize our MIFO design instead of incorporating SFT variants in Appendix F.2, with the following summarized conclusion:
> > While reducing data or lowering the learning rate can also lessen SFT-induced forgetting, such approaches risk underutilizing SFT knowledge.
>
> Regarding learning rates, our SFT learning rate is already reduced from the commonly used 1e-5 to 1e−6. This follows the standard practice in prior combined SFT+RL pipelines such as LUFFY [1], ReLIFT [2], and SRFT [3], where larger SFT learning rates are observed to harm reasoning post-training, while going below
> 1e−6 makes SFT updates too weak to be useful. The fact that these methods (and ours) must shrink the SFT learning rate to avoid degrading RL-trained capabilities is actually consistent with our view that naive SFT tends to forget RL-learned behavior: a larger SFT learning rate leads to stronger overwriting of RL-induced information. For the RL stage, we use a learning rate of 1e−6, in line with common implementations of PPO for RLHF [4] and GRPO for reasoning post-training [5]. Larger RL learning rates typically cause mode collapse, reward hacking, and unstable training due to large KL divergence, so simply “turning up” RL LR is not a safe alternative.
>
> As for using LoRA for SFT while keeping a larger learning rate or full-parameter RL: although MIFO is not primarily aimed at mitigating cross-domain forgetting, we did consider LoRA-style SFT, mentioned in Appendix B. Limitation. In practice, we do not find SFT-LoRA to be a prioritized solution because its performance gains are limited by the small number of tunable parameters. We ran experiments where we replaced full-parameter SFT with LoRA SFT (r=64,α=128) for multiple SFT+RL methods. As shown in Table Q, the LoRA-adapted variants consistently underperform their fully fine-tuned counterparts, even though LoRA naturally reduces forgetting by separating parameter subsets. Our interpretation is that, compared with our freezing strategy, LoRA exposes far fewer effective degrees of freedom to SFT, so the model cannot learn the SFT data as comprehensively. Thus, while smaller SFT LR and LoRA are reasonable baselines, their limitations further motivate our approach of explicitly identifying and protecting RL-critical parameters via MIFO, rather than relying solely on generic tricks like lowering LR or swapping in LoRA.
>
>
>
> Table Q. The comparison of methods' pass@1 test accuracies w. and w.o. lora, 1 epoch training for 15k data with Qwen2.5-1.5B-Math-Instruct.
> | Methods  | SFT-RL | ReLIFT | MIFO |
> |----------|--------|--------|------|
> | w. lora  | 26.9   | 28.1   | 28.6 |
> | w.o.lora | 27.9   | 29.5   | 30.7 |
> | $\Delta$ | +1.0    | +1.4    | +2.1  |
>
> ---
> **References**
>
> [1] Learning to Reason under Off-Policy Guidance, NeurIPS'25
>
> [2] Learning What Reinforcement Learning Can't: Interleaved Online Fine-Tuning for Hardest Questions, arxiv'25
>
> [3] SRFT: A Single-Stage Method with Supervised and Reinforcement Fine-Tuning for Reasoning, arxiv'25
>
> [4] Llama 2 Technical Report, arxiv'23
>
> [5] DeepSeekMath: Pushing the Limits of Mathematical Reasoning in Open Language Models, arxiv' 24

---

### Author Response · Authors · 2025-12-04
**Rebuttal Summary by Authors [2/2]**

**3. The Summary of Our Contribution**

- **Identify the forgetting in SFT+RL post-training pipelines.** In this study, we first identify that the forgetting phenomenon happens in reasoning post-training. Previous popular reasoning post-training pipelines combine the SFT and RL together to achieve better performance: DeepSeekMath[4] has 2 SFT$\to$RL stages; ReLIFT[2] has multiple RL$\to$SFT stages; LUFFY[1] and SRFT[3] combine SFT and RL into 1 optimisation objective. However, we first identify that this combined process have the risk of catastrophic forgetting: SFT will update many more parameter magnitudes, which will forget what RL learned smaller updates previously, causing a performance drop in reasoning post-training.

- **Our method solves reasoning forgetting during post-training.** Our method MIFO solves reasoning forgetting by two novel designs with limiting SFT parameter updates, thereby preventing the induction of forgetting on RL-learned information. **First,** we interleave SFT+RL training, and select a subset of SFT data and high-entropy tokens to decrease SFT updates on parameter magnitudes. **Second,** we freeze key RL updates on the parameter during SFT to avoid forgetting caused by parameter overwrites.

- **Our plug-and-play method achieves SOTA reasoning performance, w. data and response efficiency.** Our method achieves SOTA reasoning performance using only **1.5\%** of the SFT data and **20.4\%** of the RL data used by prior SOTA. Also, MIFO achieves a very good response efficiency with many fewer tokens. Our plug-and-play MIFO can also be integrated with different SFT+RL algorithms, unlike previous SOTA methods.

---
**We have incorporated all reviewers' suggestions faithfully into the updated paper.** We thank reviewers, AC, SAC, and PC for your valuable time and consideration!

Best regards,
Authors

---
**References**

[1] Learning to Reason under Off-Policy Guidance, NeurIPS'25

[2] Learning What Reinforcement Learning Can't: Interleaved Online Fine-Tuning for Hardest Questions, arxiv'25

[3] SRFT: A Single-Stage Method with Supervised and Reinforcement Fine-Tuning for Reasoning, arxiv'25

[4] DeepSeekMath: Pushing the Limits of Mathematical Reasoning in Open Language Models, arxiv' 24

---

### Author Response · Authors · 2025-12-04
**Rebuttal Summary by Authors [1/2]**

Dear Reviewers, AC, SAC, and PC:

Thank you for your time, effort, and commitment to ensuring a high-quality review process. We respectfully provide this summary for the ACs to record both the addressed concerns and the reviewers’ acknowledgements.

---
**1. The Summary of Rating and Reviewers' Response**

Two reviewers (p9ox,Qgmg) responded to and acknowledged our rebuttal and stated that they will raise their scores accordingly. Reviewers RRa6 and jiz5 didn't further provide their points due to the early-terminated discussion. According to the response from Reviewer p9ox and Qgmg, we at least have the **updated ratings 6644**.

|Reviewer|RRa6|p9ox|jiz5|Qgmg|
|---|---|---|---|---|
|**Final Rating**|?|**6**|?|**4**|
|Initial Rating|6|4|4|2|
|Response|1st round concerns addressed, 2nd round not responsed|Concerns addressed|Not responsed| Concerns addressed|
|Link|[Response](https://openreview.net/forum?id=HHx5KFnj8P&noteId=Y77jocb2Pi)|[Response](https://openreview.net/forum?id=HHx5KFnj8P&noteId=zFDXWliiPy)||[Response](https://openreview.net/forum?id=HHx5KFnj8P&noteId=yee1wRLOYf)|

---

**2. The Summary of Response to Reviewers**

We appreciate all of the reviewers' feedback and constructive suggestions. We are pleased that you acknowledged the contributions of our work: **The proposed method is well motivated and novel** ("an interesting view of SFT and RL in post-training", "provided good motivation for the design", Reviewer RRa6; "unique relative to prior work", " significantly different from approaches", "demonstrating the redundancy in SFT relative to RL is particularly insightful", "serves as a strong basis for the working of MIFO", Reviewer Qgmg). **SOTA performance** ("provides a promising advantage to improve the training", Reviewer RRa6; "outperforms multiple baselines", Reviewer p9ox, "achieves perfect results", "Consistently gains across different reasoning benchmarks", Reviewer jiz5). **Strong efficiency in data and response** ("good data efficiency and token efficiency", Reviewer RRa6; "improves data efficiency than baselines", Reviewer p9ox. "significantly improved data usage compared to SOTA.", Reviewer Qgmg) **Solid experiments and ablation study** ("The comprehensive experiments", "Solid ablations of complementary effects", Reviewer jiz5; "ablation study indicates MIFO is well effective", Reviewer RRa6)


Many questions are related to **more detailed implementation of our method and interpretation of observations** (Q1,3,4,5,6,8,9 of Reviewer RRa6, Q3 of Reviewer p9ox, Q1,3,4,6 of Reviewer jiz5; Q2,3,4 of Reviewer Qgmg), and **typos** (Q7 of Reviewer RRa6, Q4 of Reviewer p9ox, Q5 of Reviewer Qgmg), which we have addressed in our response and in the revised paper.

We further conduct additional experiments to answer the questions and support our claims:
- **Compare with extra models and reasoning domains** (Q2 of Reviewer RRa6, Q1 of Reviewer p9ox, Q2 of Reviewer jiz5, Q6 of Reviewer Qgmg). These experiments show that our method consistently outperforms baselines on **Coding and scientific/knowledge-based reasoning domain**, as well as the additional **Llama3.2 8B** model.

- More comprehensive experiements to justify our claims (Q10 of Reviewer RRa6, Q5 of Reviewer jiz5, Q1 of Reviewer Qgmg). These experiments show that **(i) our method can easily outperform new baselines (diff. LR/LoRA) mentioned by reviewers; (ii) Quantitatively measured forgetting does exist. (iii) Our plug-and-play method is algorithm-agnostic.**

---

### Meta-Review · Area_Chair_MGgd · 2026-01-07

**Summary:**

This paper presents MIFO, a novel post-training framework that interleaves Supervised Fine-Tuning (SFT) and Reinforcement Learning (RL) to improve reasoning capabilities of Large Language Models (LLMs) while mitigating catastrophic forgetting. The method identifies and addresses the often-overlooked phenomenon wherein SFT overwrites beneficial RL-induced updates, even within the same domain. MIFO introduces two key mechanisms to counteract this: (i) entropy-based token selection to limit high-magnitude SFT updates, and (ii) dynamic parameter freezing of RL-critical parameters during SFT. Experiments across multiple reasoning domains (math, code, scientific knowledge) and model sizes demonstrate strong generalization and substantial gains over competitive baselines, including ReLIFT, LUFFY, and SRFT.

The main concerns across reviews centered on (i) limited initial evidence of generalization beyond math/Qwen models, (ii) whether the notion of “catastrophic forgetting” was sufficiently motivated and quantitatively demonstrated, (iii) the incremental nature of the proposed techniques relative to prior interleaved SFT+RL methods, and (iv) clarity regarding computational overhead, theory–practice connection, and training dynamics. While the paper is technically competent and empirically strong, its conceptual novelty, scope of the forgetting problem, and theoretical grounding are not sufficiently compelling for acceptance at ICLR. The work is closer to a solid systems/engineering refinement of existing SFT+RL pipelines than a clear algorithmic or conceptual advance, and several key claims are only partially convincing even after rebuttal.

**Reviewer Concerns:**

**Concerns largely addressed:**

Generalizability: Initially limited to math and Qwen models, but convincingly addressed via new experiments on code generation, scientific/knowledge benchmarks, and a different backbone (Llama 3.2 8B).

Evidence for forgetting: The rebuttal added targeted experiments that explicitly quantify RL performance degradation after SFT and showed MIFO mitigating this effect.

Compute and efficiency: GPU-hour comparisons and discussion clarified that MIFO’s overhead is comparable to existing SFT+RL baselines.

Algorithm-agnostic claim: Additional experiments with alternative SFT/RL combinations supported the structural generality claim.

Presentation and clarity issues: Typos, figure inconsistencies, and missing explanations were corrected.

**Concerns partially remaining:**

Novelty vs. incremental improvement: While the authors argue that MIFO’s framing of RL→SFT forgetting is new, some reviewers may still view entropy-based selection and parameter freezing as evolutionary rather than fundamentally novel.

Theoretical analysis connection: Although clarified, the Decision–Redundancy analysis remains abstract and may still feel loosely coupled to practice for some readers.

**Reviewer Scores:**

Reviewer RRa6: Likely remains 6, possibly slightly more positive after extensive clarifications and added experiments.

Reviewer p9ox: Explicitly indicated a score increase; likely moves from 4 → 6.

Reviewer jiz5: Concerns on novelty and forgetting were substantially addressed; likely moves from 4 → 6.

Reviewer Qgmg: Increased score by 2 points after rebuttal; likely moves from 2 → 4.

---

### Decision · Program_Chairs · 2026-01-26

Reject